# Complementary control of sensory adaptation by two types of cortical interneurons

**Ryan G Natan[1], John J Briguglio[1], Laetitia Mwilambwe-Tshilobo[1], Sara I Jones[1], Mark Aizenberg[1], Ethan M Goldberg[2,3], Maria Neimark Geffen[1]\***

[1]Department of Otorhinolaryngology Head and Neck Surgery, Perelman School of Medicine, University of Pennsylvania, Philadelphia, United States; [2]Department of Neurology, University of Pennsylvania, Philadelphia, United States; [3]Division of Neurology, The Children's Hospital of Philadelphia, Philadelphia, United States

**Abstract** Reliably detecting unexpected sounds is important for environmental awareness and survival. By selectively reducing responses to frequently, but not rarely, occurring sounds, auditory cortical neurons are thought to enhance the brain's ability to detect unexpected events through stimulus-specific adaptation (SSA). The majority of neurons in the primary auditory cortex exhibit SSA, yet little is known about the underlying cortical circuits. We found that two types of cortical interneurons differentially amplify SSA in putative excitatory neurons. Parvalbumin-positive interneurons (PVs) amplify SSA by providing non-specific inhibition: optogenetic suppression of PVs led to an equal increase in responses to frequent and rare tones. In contrast, somatostatin-positive interneurons (SOMs) selectively reduce excitatory responses to frequent tones: suppression of SOMs led to an increase in responses to frequent, but not to rare tones. A mutually coupled excitatory-inhibitory network model accounts for distinct mechanisms by which cortical inhibitory neurons enhance the brain's sensitivity to unexpected sounds.

**\*For correspondence:** mgeffen@ med.upenn.edu

**Competing interests:** The authors declare that no competing interests exist.

## Introduction

Across sensory modalities, cortical neurons exhibit adaptation, attenuating their responses to redundant stimuli (*Das and Gilbert, 1999*; *Ulanovsky et al., 2003*; *Garcia-Lazaro et al., 2007*; *Asari and Zador, 2009*; *Khatri et al., 2009*). Adaptation to stimulus context is thought to increase efficiency of sensory coding under the constraints of limited resources (*Barlow, 1961*). Yet, the neuronal-circuit mechanisms that facilitate adaptation in the cortex remain poorly understood. In the primary auditory cortex (A1), the vast majority of neurons exhibit stimulus-specific adaptation (SSA, *Figure 1*). When presented with a sequence of two tones, one of which occurs frequently (termed 'standard') and another rarely (termed 'deviant'), the neuron's response to the standard tone becomes weaker, but the response to the deviant tone remains strong (*Ulanovsky et al., 2003*; *Szymanski et al., 2009*; *Farley et al., 2010*; *Fishman and Steinschneider, 2012*). Whereas SSA has also been found in sub-cortical structures e.g. in the auditory midbrain (*Malmierca et al., 2009*; *Zhao et al., 2011*; *Thomas et al., 2012*) and the auditory thalamus (*Kraus et al., 1994*; *Anderson et al., 2009*; *Antunes et al., 2010*; *Bauerle et al., 2011*), it is weak in the lemniscal areas of the auditory pathway, which project to A1, and stronger in those non-lemniscal areas which receive feedback from A1 (*Ulanovsky et al., 2004*, *Perez-Gonzalez et al., 2005*, *Duque et al., 2012*). Therefore, cortical circuits are proposed to contribute to and amplify SSA in A1 (*Ulanovsky et al., 2003*, *Szymanski et al., 2009*, *Bauerle et al., 2011*, *Fishman and Steinschneider, 2012*, *Escera and Malmierca, 2014*), through a combination of plastic modulation of thalamocortical inputs and intra-cortical inhibitory circuits, which would allow for

**eLife digest** In everyday life, we are often exposed to a mix of different sounds. An essential task for our brain is to separate the important sounds from the unimportant ones. For example, stepping out onto a busy street, you may at first be very aware of the noise of traffic. Later, you may start to ignore the din and instead only notice sounds that break the monotony: a honking car horn or maybe a stranger's voice. This is because the neurons in the auditory pathway respond differently to common and rare sounds. In particular, excitatory neurons in the region termed the 'auditory cortex' send fewer nerve impulses in response to frequent sounds, but respond vigorously to rare sounds. This phenomenon is called 'stimulus-specific adaptation', but it is not known exactly which neurons in this brain region enable this process to occur.

Now, Natan et al. have combined different cutting-edge neuroscience techniques to identify the circuit of brain cells that drives this stimulus specific adaptation. A technique called optogenetics was used to effectively 'turn off' each of two kinds of inhibitory neuron in the auditory cortex of mice, by exposing the brain to colored light from a laser.

Natan et al. found that both kinds of inhibitory neuron amplified stimulus-specific adaptation, but via different mechanisms. One of these neuron types, called 'parvalbumin-positive interneurons', exerted a general effect on excitatory neurons and suppressed responses to both frequent and rare sounds As the responses to rare sounds started off greater than the responses to frequent sounds, suppressing both by an equal amount actually led to an increase in the relative difference between them. On the other hand, the second kind of inhibitory neuron, called 'somatostatin-positive interneurons', only reduced the excitatory neurons' responses to frequent sounds; these neurons had no effect on responses to rare noises.

Future studies will test how specific adaptation in different contexts can help us to behaviorally detect rare sounds while ignoring common ones, and search for the circuits beyond the auditory cortex that support hearing in complex sound environments.

selective suppression of neuronal responses to specific stimuli (*Nelken, 2014*). Our study tests whether and how inhibitory neurons contribute to cortical SSA.

Auditory cortex, like other sensory cortices, contains morphologically and physiologically diverse inhibitory interneurons, which form dense interconnected networks with excitatory neurons (*DeFelipe, 2002*; *Douglas and Martin, 2004*). While different interneuron types have been hypothesized to carry out specialized complementary functions in sensory processing (*DeFelipe, 2002*; *Markram et al., 2004*; *Isaacson and Scanziani, 2011*; *Kepecs and Fishell, 2014*; *Marlin et al., 2015*), their function in driving changes in dynamic auditory processing has not been previously established. We hypothesized that the two most common types of interneurons in the cortex, parvalbumin- (PVs) and somatostatin-positive interneurons (SOMs) (*Xu et al., 2010*; *Rudy et al., 2011*), facilitate SSA in excitatory neurons of A1 in a complementary fashion. PVs, a subset of which receive direct thalamic inputs (*Staiger et al., 1996*), may amplify SSA in excitatory neurons by providing a constant inhibitory drive; equally strong inhibitory drive would attenuate the weak response to standard tones relatively more than the strong response to deviant tones, leading to a greater differential between standard vs deviant tone spiking response. SOMs, which target distal dendrites of pyramidal cells (*McGarry et al., 2010*; *Gentet et al., 2012*), have excitatory synapses that exhibit facilitation upon repetitive stimulation (*Reyes et al., 1998*; *Silberberg and Markram, 2007*). Therefore, inputs from SOMs may exert a stimulus-specific increase in suppression of excitatory neurons that is selective to the standard tone and does not generalize to the deviant tone. Alternatively, they may contribute to selective adaptation in excitatory neurons through differential post-synaptic integration.

To tease apart the function of different inhibitory types in SSA, we tested whether optogenetic suppression of either PV or SOM interneurons during sound presentation reduced SSA in putative excitatory neurons in the auditory cortex (*Hamilton et al., 2013*; *Pi et al., 2013*; *Weible et al., 2014*). We found that both types of interneurons contribute to SSA in the cortex, with PVs providing constant inhibition, and SOMs increasing their effect with repeated tones.

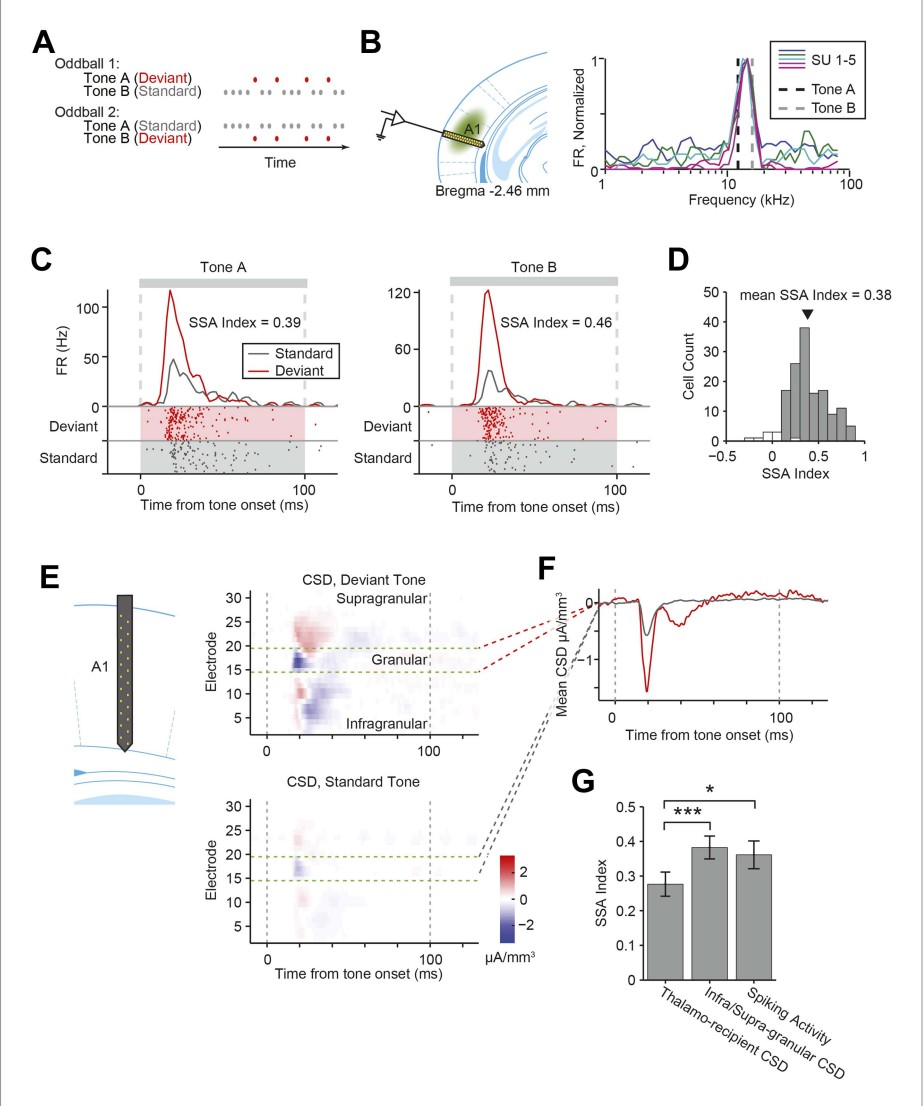

**Figure 1**. Nearly all recorded A1 neurons exhibit stimulus-specific adaptation. (**A**) Diagrams of oddball stimuli; oddball stimuli are composed of a 2.5-Hz train of 100-ms long sine-wave tone pips separated by 300 ms of silence (gray and red dots). Each tone pip is at one of two frequencies, tone A or B. In oddball stimulus 1, 10% of all pips are tone A and 90% of pips are tone B. In oddball stimulus 2, the tone probabilities are reversed. The less frequent tone is referred to as the deviant tone (red dots). The more frequent tone is referred to as the standard (gray dots). (**B**) Left: diagram of recording. Electrode was lowered perpendicular to the brain surface. Virus was injected in A1. Right: the frequencies of tones A and B (dashed black and gray lines) are selected based on the frequency response functions of neurons of interest. Mean firing rate (FR) of five co-tuned neurons (colored lines) recorded simultaneously in a single session in response to 65 dB tone pips at 50 frequencies logarithmically spaced from 1 to 80 kHz. FR is normalized to the peak response of each neuron. (**C**) A representative neuron exhibited suppressed responses to a tone presented as a standard (gray raster and PSTH) compared to the same tone presented as a deviant (red raster and PSTH). Left: responses to tone A, presented as a deviant in oddball stimulus 1, and a standard in oddball stimulus 2. Right: responses to tone B. Shaded regions indicate standard (gray) and deviant (red) tones trials. Gray dashed lines indicate tone onset and offset times. (**D**) Population histogram of stimulus-specific adaptation (SSA) index exhibited by all neurons included in the analysis. Gray and white bars indicate neurons expressing significant and non-significant SSA, respectively. Spike count for response to deviant tones was significantly greater than for response to standard tones (Wilcoxon rank sum test, one tail, p < 0.05). The black marker indicates the population average SSA index. (**E**) Left: diagram of electrode spanning A1. Right: representative peri-stimulus current source density (CSD). Top: mean response to deviant tones. Bottom: mean response to standard tones. Gray dashed lines indicate tone onset and offset. Green dashed lines indicate the location of the granular layer. Negative CSD values (blue) indicate current sinks, while positive CSD values (red)

*Figure 1. continued on next page*

*Figure 1. Continued*

indicate current sources. (**F**) Mean CSD collected from the thalamo-recipient layer, in response to standard (gray) and deviant (red) tones. Gray dashed lines indicate tone onset and offset. (**G**) Mean SSA index across sessions measured from thalamo-recipient granular layer CSD, infra- and supra-granular layer cortical CSD and mean neuronal spiking activity SSA index averaged over sessions.

The following figure supplement is available for figure 1:

**Figure supplement 1**. Local field potentials recorded in A1 exhibit SSA.

## Results

### Nearly all neurons in A1 exhibit SSA

We recorded spiking activity of neurons as well as local field potentials (LFPs) in A1 in head-fixed mice under light isoflurane anesthesia. SSA was measured from the firing rate (FR) of neurons in response to tones presented as a series of 'oddball' stimuli. Each oddball stimulus consisted of a sequence of tone pips at one of two frequencies (tones A and B). In each oddball stimulus, one tone was presented as the rare (deviant) tone, while the other was presented as the frequent (standard) tone (A to B ratio of 90:10 or 10:90, *Figure 1A*). A third stimulus was also presented (equal stimulus), with tones A and B being presented equally often (50:50). The frequencies of tone A and B were selected at 0.39 octave intervals, narrower than the typical tuning bandwidth of A1 neurons (*Hackett et al., 2011*; *Guo et al., 2012*; *Kanold et al., 2014*), such that they activated the majority of recorded neurons on each session (*Figure 1B*).

As expected, for a representative neuron recorded in A1, the mean FR in response to a tone was lower when the tone was presented as the standard than as the deviant (*Figure 1C*), exhibiting SSA. To quantify the level of adaptation for each neuron, we computed the index of the change in FR to the same tone when it was presented as the deviant vs the standard (SSA index). SSA index is 1 when adaptation is complete (i.e., no response to the standard, and significant response to the deviant), and 0 when there is no adaptation (i.e., the response to the standard and deviant is equal). Almost all neurons recorded in A1 exhibited significant SSA (*Figure 1D*, standard tone-evoked FR significantly lower than the deviant tone-evoked FR in N = 138 out of 147 neurons, Wilcoxon rank sum test p < 0.05).

### Contribution of thalamocortical inputs to SSA

We first tested whether SSA is present in inputs from the thalamus. Current source density (CSD) analysis has been extensively used to quantify inputs from the thalamus (*Metherate and Cruikshank, 1999*; *Kaur et al., 2005*; *Szymanski et al., 2009*; *Happel et al., 2014*). We used a linear probe to record LFPs using electrodes spaced 50 microns apart inserted perpendicularly to brain surface in the primary auditory cortex. The multi-electrode probe is 775-μm long, spanning layers 1–6 of mouse A1. CSD is computed as the second spatial derivative of the LFPs across the depth of the cortex (*Figure 1E*, *Figure 1—figure supplement 1A*, 20 sessions, 15 mice). Typically, in response to tones, CSD exhibits a negative basin, termed sink, within a short delay of tone onset, localized to electrodes in thalamo-recipient layer (*Figure 1F*, *Figure 1—figure supplement 1B*) (*Kaur et al., 2005*; *Szymanski et al., 2009*). The amplitude of current in the sink was taken as a measure of the combined strength of post-synaptic inputs onto layer 4 neurons, which should reflect the strength of the thalamic inputs to the cortex (*Metherate and Cruikshank, 1999*; *Kaur et al., 2005*; *Szymanski et al., 2009*; *Happel et al., 2014*).

We compared the amplitude of the CSD sink for each tone when presented as a deviant or standard, and computed their ratio (*Figure 1F*). The sink amplitude was lower for the standard as compared to the deviant tones (*Figure 1F,G*), suggesting that excitatory signals produced by thalamo-cortical inputs exhibit SSA, consistent with previous findings (*Szymanski et al., 2009*). This finding supports the 'adaptation in narrowly tuned inputs' model, which postulates that SSA in broadly tuned neurons in A1 reflects adaptation in either thalamocortical inputs or at the stage of integration of thalamocortical inputs, specific to inputs tuned to the standard tone (*Mill et al., 2011*;

*Taaseh et al., 2011*; *Nelken, 2014*). Importantly, across sessions, the SSA index of the granular layer CSD sinks was significantly lower than that of either the non-thalamo-recipient layers ($\Delta = -28\%$, p-value from one-sided test after correction (p1) = $6e^{-4}$, z = $-3.4$, Bonferroni corrected for two tests (C = 2)) or the SSA index of the mean spiking activity of A1 neurons ($\Delta = 23\%$, p1 = 0.029, z = $-2.1$, C = 2) in each session (N = 20 sessions in 15 mice, *Figure 1G*), suggesting that additional intra-cortical mechanisms may contribute to SSA in the cortex.

## Suppression of either PVs or SOMs decreases SSA in putative excitatory neurons

We next tested whether cortical inhibitory interneurons may contribute to SSA. Since different inhibitory neuronal subtypes can differentially affect sensory responses of putative excitatory neurons (*Lee et al., 2012*; *Wilson et al., 2012*; *Cottam et al., 2013*), we separately tested the role of PVs and SOMs. We used targeted viral delivery in the auditory cortex of mice to drive Archaerhodopsin (Arch) expression, which hyperpolarizes neurons when stimulated by light, in either PVs or SOMs (*Chow et al., 2010*). A modified adeno-associated virus (AAV) encoding anti-sense code for Arch and a fluorescent reporter, under the FLEX cassette, was injected into PV-Cre or SOM-Cre mice (*Boyden et al., 2005*; *Sohal et al., 2009*; *Cardin et al., 2010*; *Zhang et al., 2010*; *Deisseroth, 2011*) (*Figure 2A*). 2 weeks following virus injection, Arch was expressed selectively in PVs or SOMs in auditory cortex at expected levels (*Kvitsiani et al., 2013*) (*Figure 2B,C* PV-Cre: N = 250 neurons in 4 mice, specificity = $92 \pm 1\%$, efficiency = $73 \pm 5\%$. SOM-Cre: N = 149 neurons in 5 mice, specificity = $95 \pm 2\%$, efficiency = $86 \pm 5\%$). To activate Arch, a light guide was positioned to cast 180 mW/mm 532-nm light onto A1 surface, perpendiular to cortical layers. In vitro intracellular recordings from optically identified PVs or SOMs (*Figure 2—figure supplement 1*, *Figure 2—figure supplement 2*) demonstrate that light cast over the auditory cortex in vitro drives a strong suppressive current (*Figure 2D*, *Figure 2—figure supplements 1C,D, 2C,D*) and hyperpolarizes the membrane potential in these neurons (*Figure 2—figure supplements 1B, 2B*). Assuming a 100-fold attenuation of light over 1 mm of brain tissue (*Aravanis et al., 2007*), the estimated irradiance in the deepest cortical layer (1.8 mW/mm$^2$) was strong enough to induce hyperpolarizing current in neurons in vitro (*Figure 2D*). In vivo, in both PV-Cre and SOM-Cre mice, illuminating the auditory cortex suppressed spiking activity in a small subset of recorded neurons (*Figure 2E,F*, left, putative inhibitory neurons) and increased activity in a great majority of recorded neurons (*Figure 2E,F* right, putative excitatory neurons). Shining light over A1 increased spontaneous neuronal activity in the majority of the recorded neurons in both PV-Cre mice (N = 115 neurons, 102 increased, 0 decreased, in 10 mice) (*Figure 2G*) and SOM-Cre mice (N = 104 neurons, 61 increased, 3 decreased, in 9 mice) (*Figure 2H*). These measurements demonstrate that casting light over A1 selectively and effectively suppresses the activity of either PVs or SOMs.

To test the function of PVs and SOMs in SSA, their activity was suppressed during every fifth tone of the oddball stimulus by illuminating A1 (*Figure 3A*). To directly test the effect of interneuron suppression, we computed the SSA index separately on light-on and light-off trials for neurons responsive to both tones A and B (SSA was found in 63 out of 67 tone-responsive neurons in PV-Cre mice, 42 out of 43 tone-responsive neurons in SOM-Cre mice). Photosuppression of either PVs or SOMs affected the responses of neurons to the tones (*Figure 3B,C*), resulting in a significant reduction in SSA index across the population (*Figure 3E,F*, PV-Cre: $\Delta = -41\%$, p1 = $1e^{-12}$, t(66) = 8.6. SOM-Cre: $\Delta = -25\%$, p1 = $2e^{-6}$, t(42) = 5.4). Photo-manipulation-affected responses only to the tone during which it was presented, but not to subsequent tones (*Figure 3—figure supplement 1*). Additionally, photo-manipulation was limited to cortex since it did not affect thalamo-recipient layer CSD tone responses and SSA (*Figure 3—figure supplement 2*). In a control group of PV-Cre or SOM-Cre mice (6 mice), we injected a modified AAV, which encoded anti-sense fluorescent reporter alone under the FLEX cassette, and computed the effect of casting light on SSA (SSA was found in 33 out of 37 tone-responsive neurons in control mice). In this control group, SSA was not affected by light (*Figure 3D,G*, p > 0.05, t(36) = $-2.0$), confirming that Arch expression was required for the effect of the light. Therefore, the effects of interneurons are specific to intra-cortical mechanisms. These results demonstrate that both types of interneurons contribute to the reduction of the response of the neuron to the stimulus during SSA.

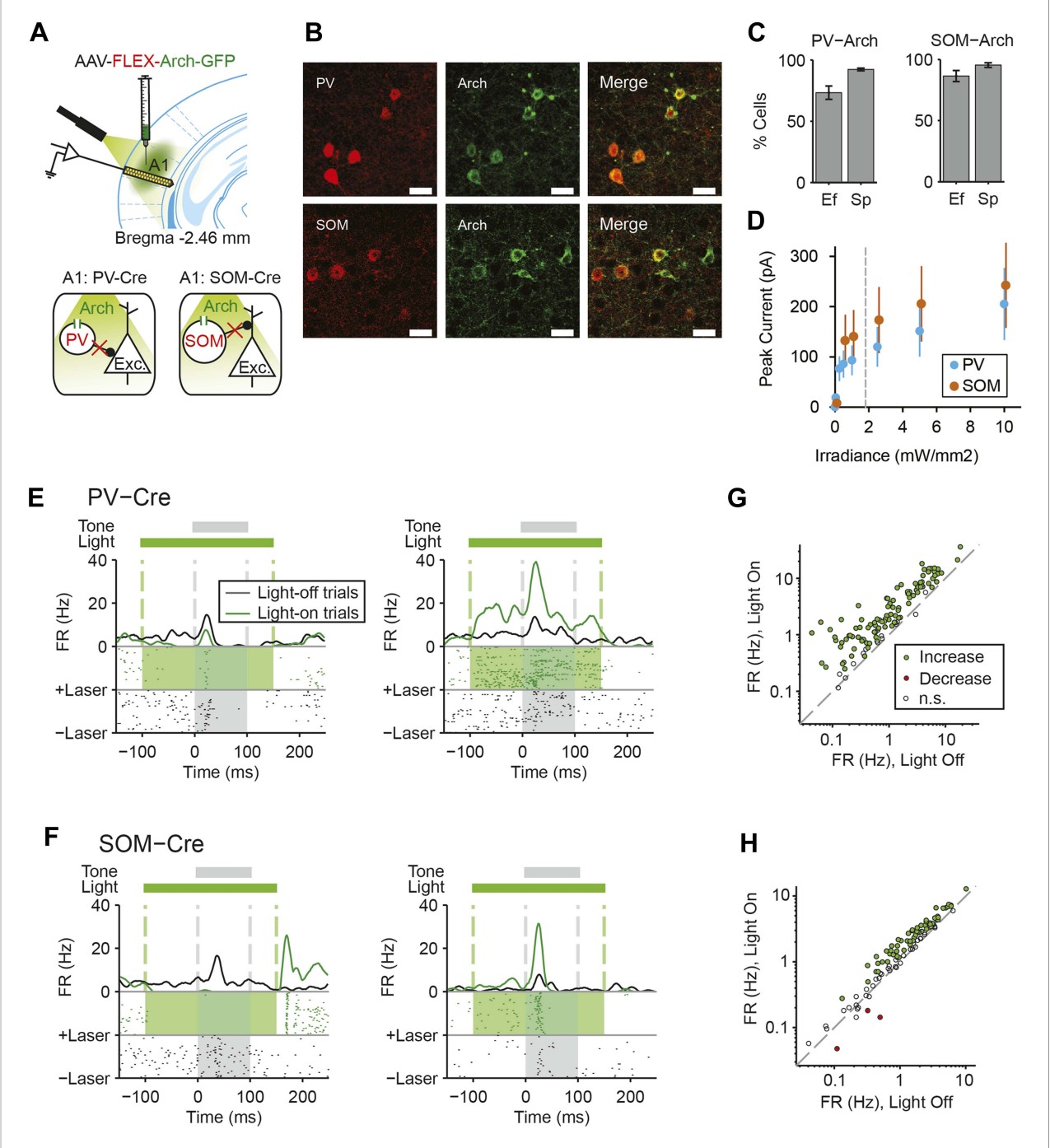

**Figure 2**. Cell type-specific optogenetic suppression of parvalbumin-positive and somatostatin-positive neurons. (**A**) Optogenetic methods diagram. Top: A1 was injected with AAV-FLEX-Arch-GFP. During experiments, an optic fiber was positioned to target A1 and neuronal activity was recorded using a multichannel silicon probe in A1. Bottom: green light (532 nm) suppresses PVs in PV-Cre mice or SOMs in SOM-Cre mice. (**B**) Transfection of interneurons with Archaerhodopsin (Arch). Immunohistochemistry demonstrating co-expression of the Arch and an interneuron-type reporter in A1. Top: PV-Cre mouse A1. Red: anti-body stain for parvalbumin. Green: Arch-GFP. Merge; co-expression of Arch and parvalbumin. Bottom: SOM-Cre mouse A1. Red: anti-body stain for somatostatin. Green: Arch-GFP. Merge; co-expression of Arch and somatostatin. Scale Bar = 25 μm. (**C**) Efficiency and specificity of transfection of interneurons with Arch. Bar Plots: efficiency (Ef) and specificity (Sp) of visual transfection of PVs (top) and SOMs (bottom) with Arch. Ef, percent of labeled interneurons expressing Arch. Sp, percent of Arch-expressing cells, which are also labeled interneurons. (**D**) Mean Arch-mediated outward current evoked in response to increasing photostimulation power, recorded in vitro by whole-cell patch recording in putative excitatory neurons from PV-Cre (blue, N = 5) and Som-Cre (orange, N = 5) mice. The gray dashed line indicates the level of irradiance expected in in vivo experiments at the deepest recording sites, in cortical layer 6. (**E**, **F**) Tone responses of representative neurons, which are suppressed (left) or activated (right) by photostimulation, from PV-Cre (**E**) and

*Figure 2. Continued*

SOM-Cre (**F**) mice. Raster plot of spike times (bottom) and PSTH (top) of a single neuron response to a 100-ms long tone (gray dashed lines, shaded region) on light-on (overlapping 250-ms light pulse, green shading) and light-off trials. Light-on trials: green. Light-off trials: black. (**G**, **H**) Modulation of spontaneous FR by interneuron photosuppression recorded in PV-Cre (**G**) and SOM-Cre (**H**) mice. Each neuron is represented by a circle that is filled for those with significantly increased (green) or decreased (red) FR or unfilled for those without significant modulation. Gray dashed line, identity line.

The following figure supplements are available for figure 2:

**Figure supplement 1**. Optogenetic control of PVs in mouse primary auditory cortex via photostimulation of Arch in acute slices.

**Figure supplement 2**. Optogenetic control of SOMs in mouse primary auditory cortex via photostimulation of Arch in acute slices.

## PVs and SOMs differentially suppress putative excitatory neuron responses to standard and deviant tones

A decrease in the SSA index may be due to several factors: (1) an increase in response to the standard only, (2) a decrease in response to the deviant, or (3) an increase in response both to the standard and the deviant, but with a relatively greater increase for the standard. Therefore, we next investigated the effect of interneuron photosuppression on FR of putative excitatory neurons evoked by the standard and deviant tones separately. The effects of PVs and SOMs diverged; in addition to increasing spontaneous activity ($\Delta$ = 185%, p-value from one-sided t-test after correction (p2) = $3e^{-11}$, t(159) = −7.2), suppressing PVs led to increased FR to both the standard ($\Delta$ = 102%, p2 = $3e^{-11}$, t(159) = −7.2) and deviant ($\Delta$ = 56%, p2 = $9e^{-12}$, t(159) = −7.4) tones (N = 160, *Figure 4A–C*, *Figure 4—figure supplement 1A*). 83% of neurons exhibited greater FR to the standard and 46% to the deviant during PV suppression (*Figure 4—figure supplement 1B*). The difference in FR due to suppression of PVs was not significantly different between the standard and deviant tones (p2 > 0.05, t(159) = −0.1, C = 2) but both were greater than the difference in the spontaneous FR (standard: $\Delta$ = 25%, p2 = 0.001, t(159) = −3.6, C = 2. Deviant: $\Delta$ = 26%, p2 = 0.039, t(159) = −2.4, C = 2), indicating that the change in tone-evoked FR was similar regardless of tone probability (*Figure 4B*, bottom panel). Because an equal increase in the FR produces a weaker *relative* effect on the response to the deviant (which is higher than to the standard), PV inactivation decreases SSA index (*Figure 3E*).

By contrast, suppressing SOMs led to an increase in FR for spontaneous activity ($\Delta$ = 46%, p2 = $2e^{-9}$, t(113) = −6.5) and during the standard ($\Delta$ = 29%, p2 = $2e^{-8}$, t(113) = −6.1) but not deviant (p2 > 0.05, t(113) = −0.8) tone (N = 114, *Figure 4D–F*, *Figure 4—figure supplement 1C*). 52% of neurons exhibited greater FR to the standard and only 11% to the deviant during PV suppression (*Figure 4—figure supplement 1D*). The increase in FR for spontaneous activity was not different than that during the standard tone (p2 > 0.05, t(113) = 0.2, C = 2) and the differences in FR due to suppression of SOMs were stronger for spontaneous activity and the standard tone than the deviant tone (spontaneous: $\Delta$ = 390%, p2 = 0.004, t(113) = 3.1. Standard: $\Delta$ = 378%, p2 = 0.005, t(113) = 3.1) (*Figure 4E*, bottom panel), thereby accounting for the change in SSA with SOM inactivation (*Figure 3F*). Responses to the equal stimulus evoked consistent, yet weaker effects (*Figure 4—figure supplement 2*).

PVs and SOMs differ in their density among different layers of the cortex and in laminar sources and targets of their inputs and outputs (*Markram et al., 2004*; *Xu and Callaway, 2009*; *Fino et al., 2013*). The effects of PV and SOM suppression on SSA had differential laminar distribution (*Figure 4—figure supplement 3*). The effect of PVs on SSA was equally strong in the supra-granular and infra-granular layers, but stronger in the granular layer, that is, the thalamo-recipient layer. This differential effect is consistent with the relative proportion of cortical interneurons that are PVs, which is higher in granular than either in infra- or supra-granular layers (*Lee et al., 2010*; *Xu et al., 2010*; *Ouellet and de Villers-Sidani, 2014*). In contrast, suppressing SOMs reduced SSA in the granular and infra-granular, but not supra-granular layers. The relative proportion of cortical interneurons that are SOMs is greatest in the granular and infra-granular layers, but still present in supra-granular layers (*Lee et al., 2010*; *Xu et al., 2010*; *Ouellet and de Villers-Sidani, 2014*). As some SOMs predominantly target the distal dendrites of pyramidal neurons (*Markram et al., 2004*), the effect of suppressing SOMs in supra-granular layers may be evident in recordings of pyramidal neurons with

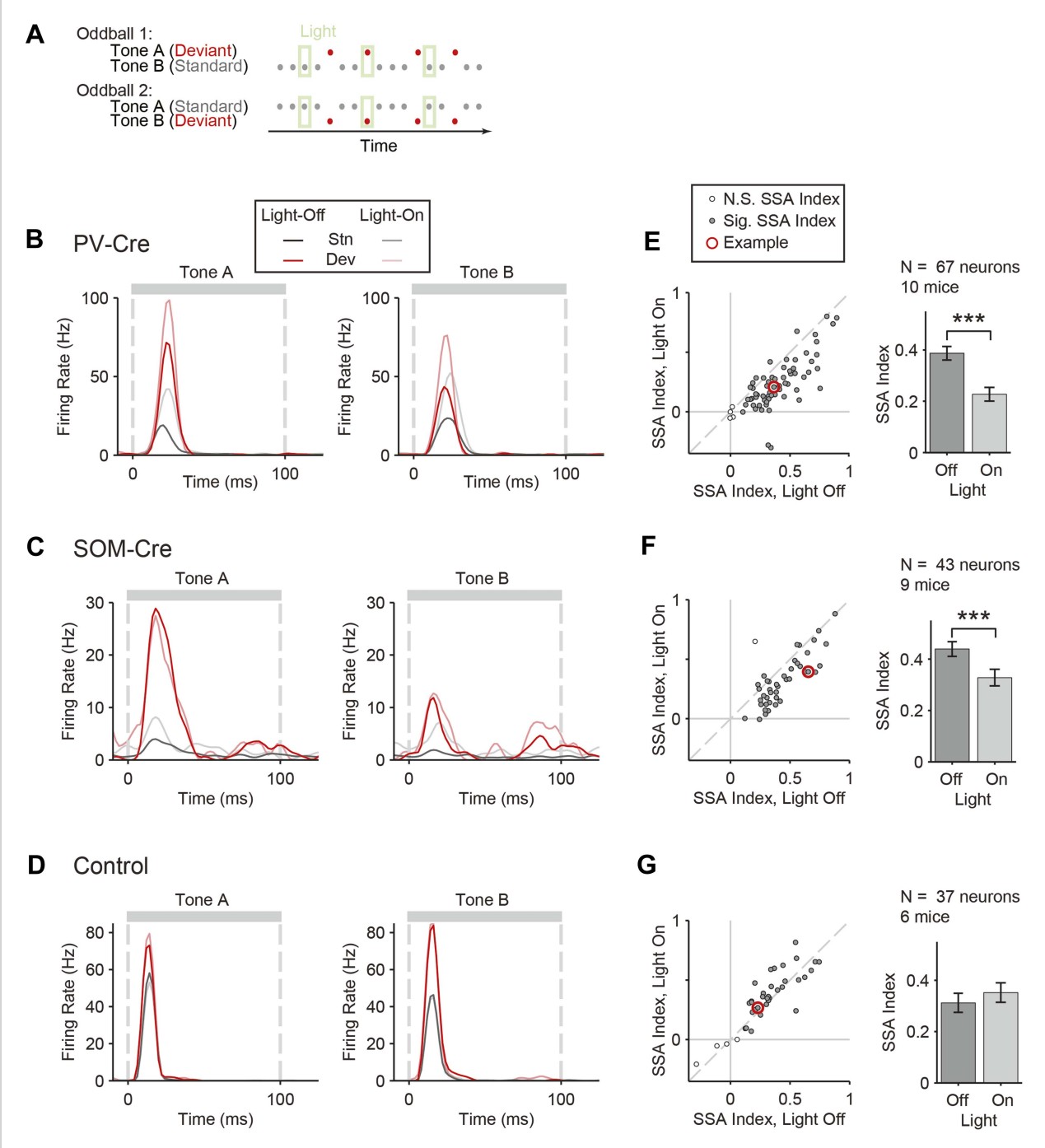

**Figure 3**. Optogenetic suppression of either PVs or SOMs reduces SSA in putative excitatory neurons in the auditory cortex. (**A**) Diagram of oddball stimuli with light; two oddball stimuli are presented (as in **Figure 1A**), with 250-ms light pulses (green bars) delivered during every fifth tone, starting 100 ms before tone onset. (**B–D**) Representative neuron PSTH in response to tone A (left) and B (right) as a standard (gray) or deviant (red) on light-on (light colors) and light-off trials (dark colors). Neurons recorded in PV-Cre (**B**, **E**), SOM-Cre (**C**, **F**), and control (**D**, **G**) mice. (**E–G**) Effect of interneuron photosuppression on SSA. Left: SSA index on light-on vs light-off trials. Each neuron is represented by a circle that is filled if the neuron exhibits significant SSA, that is, its FR in response to deviant tones is greater than that to standard tones. The respective representative neuron in **B**, **C**, and **D** is indicated by a red circle. Gray dashed line, identity line. Right: mean SSA index on light-on (green) and light-off (gray) trials over neuronal population.

*Figure 3. continued on next page*

*Figure 3. Continued*

The following figure supplements are available for figure 3:

**Figure supplement 1**. Photostimulation during standard tone does not affect SSA during subsequent tones on light-off trials.

**Figure supplement 2**. Interneuron photosuppression does not affect thalamocortical responses to standard or deviant.

---

cell bodies in deeper layers, supporting our results. In addition, cortical extracellular recordings may be biased toward neurons in granular and infra-granular layers, precluding adequate sampling of activity in superficial layers. In controls, we did not observe a difference in the effect of light on SSA across layers, demonstrating that the differences are not due to differential artifact of light stimulation.

Our results indicate that both PVs and SOMs affect SSA, but in different ways: (1) the increase in the FR of putative excitatory neurons due to PV suppression is constant, either during presentation of the standard or the deviant, and greater than changes in spontaneous activity. Thus, PVs amplify SSA in excitatory neurons by exerting a *relatively* stronger inhibitory drive for the standard than for the deviant. (2) Suppression of SOMs leads to increased putative excitatory neuron activity only during the spontaneous firing or the presentation of the standard, but not for the deviant. This suggests that the strength of SOM-mediated inhibitory drive is not significant in response to the deviant but increases with repeated presentations of the standard.

In neurons exhibiting SSA, responses to the deviant are stronger than to the standard. This difference might lead to a 'ceiling' effect, reducing the effect of PV photosuppression on FR to the deviant, but not standard (*Olsen et al., 2012*). However, restricting the analysis to two subpopulations of neurons, which have matched mean and standard deviation of FR to the standard vs the deviant tones (*Ulanovsky et al., 2004*; *Rust and Dicarlo, 2010*), preserved the observed effects of photosuppression (*Figure 4—figure supplement 4*). Suppressing PVs led to an equal increase in FR to both the standard and the deviant tone (N = 54—standard: $\Delta$ = 62%, p2 = $6e^{-8}$, t(53) = 6.3. Deviant: $\Delta$ = 55%, p2 = $3e^{-5}$, t(53) = 4.5. Standard vs deviant: p2 > 0.05, t(53) = 0.5). In contrast, suppressing SOMs led to a significant increase in FR to the standard, but no change in FR to the deviant (N = 44—standard: $\Delta$ = 30%, p2 = $7e^{-6}$, t(43) = 5.1. Deviant: p2 > 0.19, t(43) = 1.3. Standard vs deviant: $\Delta$ = 382%, p2 = $6e^{-4}$, t(43) = 3.7).

For neurons that responded more strongly to one of the tones ('strong' vs 'weak' tone), a ceiling effect would predict that the effect of interneuron suppression would be stronger for the weak than the strong tone. However, PV and SOM suppression exhibited a similar effect on responses to the strong and the weak tones in neurons that exhibited differential responses to two tones (*Figure 4—figure supplements 5, 6*). Suppressing PVs led to similar increases in tone-evoked FR between weak and strong tones for both deviant (N = 51, p2 > 0.05, t(50) = 1.0) and standard tones (p2 > 0.05, t(50) = −1.9). Suppressing SOMs also led to similar differential effects between strong and weak tones; standard tone-evoked FR increased equally (N = 34, p2 > 0.05, t(33) = 1.1) and deviant tone-evoked FR was equally unchanged (p2 = 0.05, t(33) = −0.1). Combined, these analyses demonstrate that the effect of PV photosuppression on SSA cannot be explained by the ceiling effect for either PVs or SOMs.

Although Arch drove strong currents in both SOM and PV neurons (*Figure 2D*, *Figure 2—figure supplements 1, 2*), there might be a difference in expression level or efficacy of Arch between SOM-Cre and PV-Cre mice, leading to a stronger effect of photosuppression in PV-Cre than in SOM-Cre mice on tone-evoked FRs (*Figure 4B,E*). Alternatively, the difference might be attributable to the morphological or functional differences between SOMs and PVs. To address this confound, we selected tone responses that exhibited matched difference in standard tone-evoked FR between light-on and light-off trials (N = 66, *Figure 4—figure supplement 7*). Within these matched subpopulations, PV and SOM photosuppression exhibited differential effects similar to those of the whole population. The change in FR due to PV suppression was not significantly different between responses to the standard and deviant (p2 > 0.05, t(65) = −0.3, C = 3). By contrast, the change in deviant tone-evoked FR due to SOM suppression was significantly weaker than that for the standard tone ($\Delta$ = −78%, p2 = 0.003, t(3.5), C = 3). By the design of the analysis, the effect of PV or SOM suppression on standard tone-evoked FR was nearly identical (p1 > 0.05, t(65) = −0.1, C = 3).

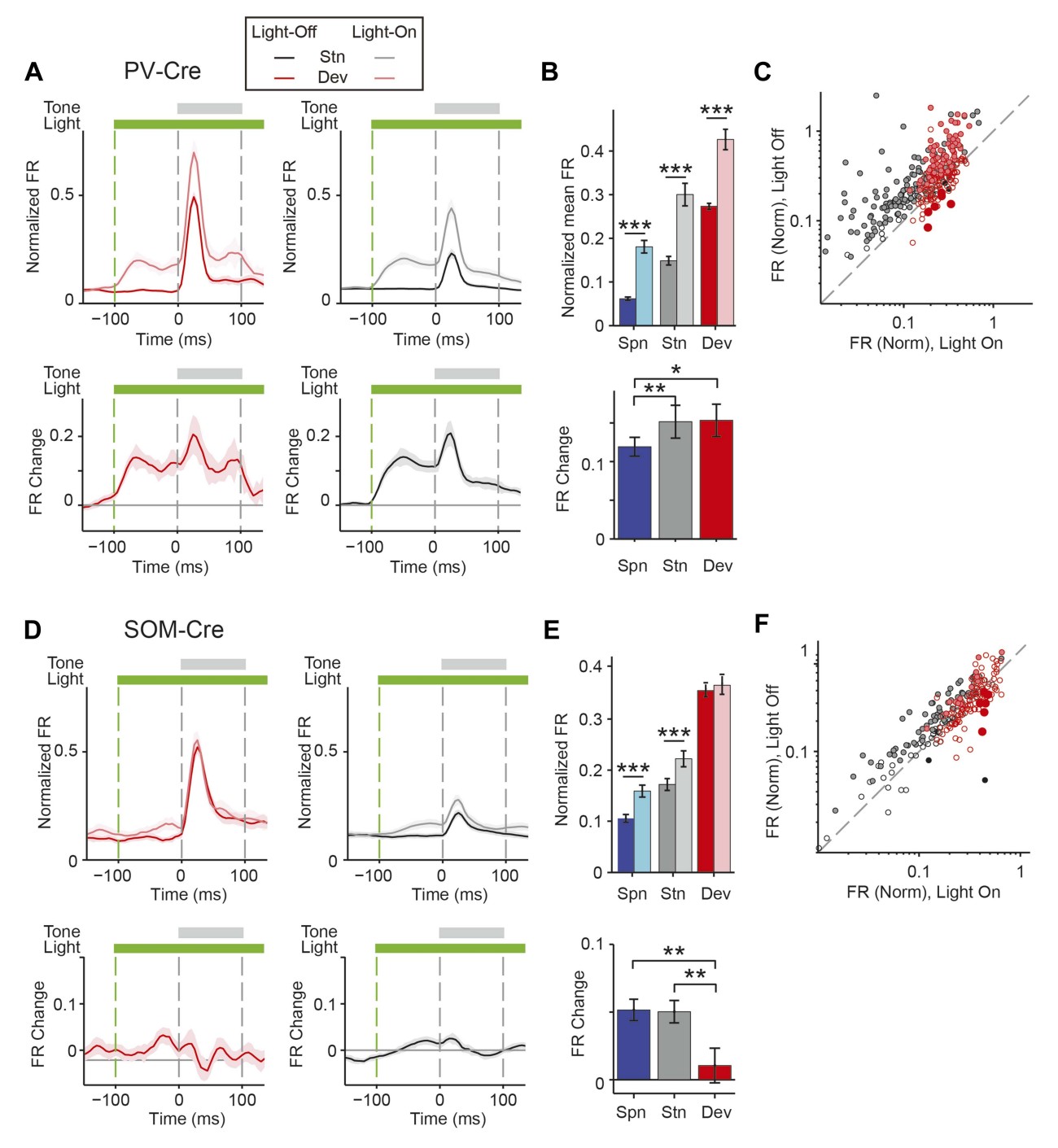

**Figure 4**. PVs and SOMs differentially affect response to standard and deviant tones. (**A**, **D**) Top: mean response to deviant (left, red) and standard (right, black) tones, during light-on (light colors) and light-off trials (dark colors). Bottom: mean of the difference between responses on light-on and light-off trials for each neuron for deviant (left, red) and standard (right, black) tone. Each trace is a population average of putative excitatory neuron PSTHs normalized to each neuron's maximum deviant tone-evoked FR on light-off trials. Shaded regions around traces indicate standard error (SE). Dashed lines indicate light onset (green) and tone onset and offset (gray). Neurons recorded in PV-Cre (**A**), SOM-Cre (**D**) mice. (**B**, **E**) (Top) Mean population FR on light-on and light-off trials; (bottom) mean population FR difference between light-on and light-off conditions for deviant (red) and standard (gray) tones and spontaneous activity (blue). Normalization as in **A**. Neurons recorded in PV-Cre (**B**), SOM-Cre (**E**) mice. (**C**, **F**) Modulation of PV-Cre mouse putative excitatory neuron FR response to tones by interneuron photosuppression. Neuronal responses to each tone are represented by two circles, one for standard (black) and one for deviant (red) tone

*Figure 4. continued on next page*

*Figure 4. Continued*

responses. Filled circles represent significantly increased (gray, pink) or decreased (black, red) response; unfilled circles: responses without significant modulation. Gray dashed line, identity line. Neurons recorded in PV-Cre (**C**), SOM-Cre (**F**) mice.

The following figure supplements are available for figure 4:

**Figure supplement 1**. PVs and SOMs differentially affect response to standard and deviant tones.

**Figure supplement 2**. Consistent effects of PV and SOM suppression in response to equal probability tones.

**Figure supplement 3**. PVs and SOMs have differential effects on SSA across different layers of cortex.

**Figure supplement 4**. Differences between PV and SOM effects on standard and deviant tones are preserved for subsets of neurons matched for FR.

**Figure supplement 5**. Effects of PV suppression are identical for tones that evoke strong or weak responses in putative excitatory neurons.

**Figure supplement 6**. Effects of SOM suppression are identical for tones that evoke strong or weak responses in putative excitatory neurons.

**Figure supplement 7**. Differences between PV and SOM effects on standard and deviant tones are preserved for subsets of neurons matched for strength of laser effects on standard tones.

**Figure supplement 8**. Differences between PV and SOM effects on standard and deviant tone responses are preserved when FRs are normalized by the mean onset response.

However, the change in deviant tone-evoked FR was greater for PV photosuppression than SOM photosuppression ($\Delta$ = 404%, p1 = 0.029, t(65) = 2.4, C = 3). Since the observed differential effects of PV and SOM suppression persisted in subsets of neurons that were matched for photosuppression-induced change in standard tone-evoked FR, these differences are unlikely due to differential expression or efficacy of Arch in the PV-Cre and SOM-Cre mice, but rather reflect functional differences between the two types of interneurons.

## SOM-mediated suppression of putative excitatory neurons increases with repeated presentations of the standard tone, whereas PV-mediated suppression remains stable

Within the oddball sequence, after the presentation of the deviant tone, SSA takes several repeats of the standard tone to reach an adapted state (*Ulanovsky et al., 2004*). Consistent with previous findings (*Ulanovsky et al., 2004*), presentation of the deviant tone temporarily reduced SSA without photosuppression (*Figure 5A–C*, dark color bars); following the deviant tone ($T_0$), the first two standard tones ($T_1$ and $T_2$) evoked elevated FRs compared to the fourth standard tone ($T_4$) (PV-Cre, *Figure 5B*—N = 148, $T_1$: $\Delta$ = 60%, p2 = 3e$^{-8}$, t(146) = 6.3, C = 11, $T_2$: $\Delta$ = 26%, p2 = 0.043, t(146) = 2.9, C = 11. SOM-Cre, *Figure 5C*—N = 102, $T_1$: $\Delta$ = 72%, p2 = 1e$^{-5}$, t(101) = 5.2, C = 11, $T_2$: $\Delta$ = 31%, p2 = 0.013, t(101) = 3.3, C = 11). The third standard tone ($T_3$) and the tone prior to the deviant tone ($T_{-1}$) evoked responses similar to $T_4$ (PV-Cre, *Figure 5B*—$T_{-1}$ and $T_3$: p2 > 0.05, t(146) < 2.5, C = 11. SOM-Cre, *Figure 5C*—$T_{-1}$ and $T_3$: p2 > 0.05, t(101) < 2.9, C = 11). Neurons in which response to $T_0$ did not produce spikes were excluded. Suppressing PVs led to a significant and equal increase in FR to four consecutive presentations of the standard following the deviant (*Figure 5B*, left, for each tone, $T_{-1}$ through $T_4$, with light-on compared to $T_4$ with light-off: $\Delta$ > 132%, p2 < 2e$^{-9}$, t(146) ≥ 6.8, C = 11. *Figure 5B*, right, change in FR between light-on and light-off responses to each $T_{-1}$ through $T_3$ as compared to $T_4$: p > 0.05, t(146) < 1.8, C = 5). In contrast with PVs, suppressing SOMs led to a progressively increasing effect on FR to consecutive presentations of the standard tone following the deviant (*Figure 5C*, left, for each standard tone, $T_{-1}$ through $T_4$, with light-on compared to $T_4$ with light-off: $\Delta$ > 64%, p2 < 9e$^{-4}$, t(101) ≥ 4.1, C = 11. *Figure 5C*, right, difference between FR change in $T_1$ and $T_4$ with light-on: p = 0.008, t(101) = −3.2, C = 5. Repeated measures ANOVA with tone number ($T_1$ through $T_4$) as a factor: F(3, 300) = 4.30, p = 0.0054). These results are consistent with the

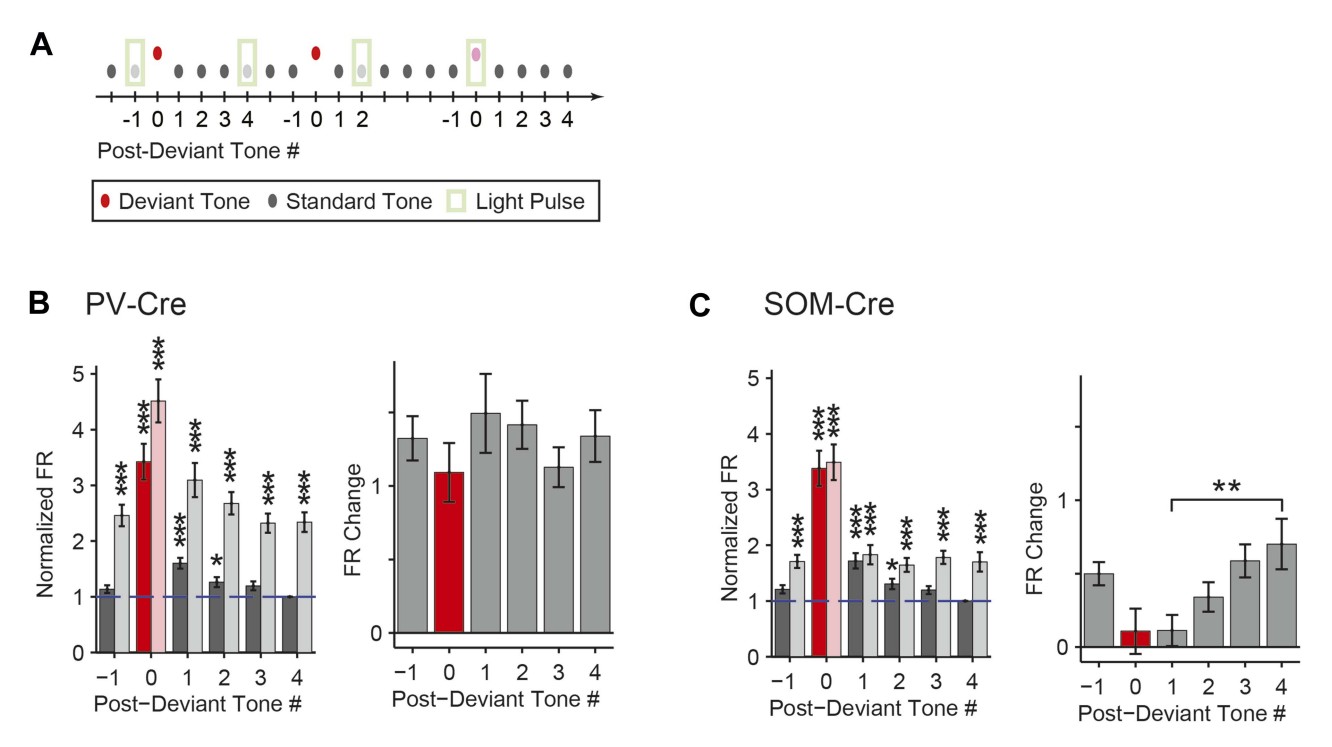

**Figure 5**. Post-deviant time course of interneuron-mediated effect on SSA. (**A**) Diagram of oddball stimuli illustrating post-deviant tone number used in subsequent analysis; Tones and light pulses are as indicated in *Figure 3A*. Numbers indicate each tone position relative to deviant tones. Responses to any standard tones following light-on standards were excluded from the analysis. (**B**, **C**) Left: mean population FR in response to standard tones (gray) subsequent to deviant tones (red) within the oddball sequence on light-off (dark colors) and light-on (light colors) trials. All responses are normalized to the response to the fourth post-deviant standard tone on light-off trials (green dashed line). Right: difference between FR on light-on and light-off trials in response to standard (gray) and deviant (red) tones. (**B**): PV-Cre mice. (**C**): SOM-Cre mice.

The following figure supplement is available for figure 5:

**Figure supplement 1**. Initial time course of interneuron-mediated effect on SSA.

interpretation that the inhibitory drive from PVs is constant throughout the stimulus regardless of tone history, whereas the effect of SOM modulation increases with repeated presentations of the standard tone.

The time course of the effect of interneuron photosuppression on FR of the putative excitatory neurons at the beginning of each oddball sequence exhibited similar differences between PVs and SOMs. After the onset of each oddball sequence, SSA develops over the course of several standard tone presentations (*Ulanovsky, 2004*). As expected, on light-off trials, FR decreased in response to the standard tone over the first 20 repetitions of the tone (*Figure 5—figure supplement 1*). For PV-Cre mice, the difference in FR to the standard tone between light-on and light-off trials did not change over this time and stayed positive for the remainder of the oddball stimulus (*Figure 5—figure supplement 1A*). Over the first 20 trials, FR adapted with a similar time course for both the light-on and light-off trials, so the change due to PV photosuppression in FR to standard stayed constant (*Figure 5—figure supplement 1A,B*). In contrast, for SOM-Cre mice, FR on light-on trials increased over the first 40 trials, whereas on light-off trials, it decreased (*Figure 5—figure supplement 1A*). As a result, the difference due to photo-manipulation in FR to the standard tone increased over the first 40 trials and then stayed consistently positive throughout the stimulus presentation (*Figure 5—figure supplement 1C*). These results demonstrate that the PV-mediated effect on putative excitatory neuronal responses did not change with repeated presentations of the standard tone, whereas the SOM-mediated effect increased with the repeated stimulus.

## PVs and SOMs exhibit SSA

In order to understand how PVs and SOMs exert differential control of SSA in putative excitatory neurons, we used optogenetic tagging to identify the specific interneurons and to quantify whether PVs and SOMs exhibited SSA (*Lima et al., 2009*). Through targeted viral delivery to AC, we drove Channelrhodopsin-2 (ChR2) expression, which depolarizes neurons when stimulated by light, in either PVs or SOMs (*Chow et al., 2010*) (*Figure 6A,D*, *Figure 6—figure supplement 1A*). A modified AAV encoding anti-sense code for ChR2 and a fluorescent reporter, under the FLEX cassette, was injected into PV-Cre or SOM-Cre mice (*Boyden et al., 2005*; *Sohal et al., 2009*; *Cardin et al., 2010*; *Zhang et al., 2010*; *Deisseroth, 2011*) and resulted in specific expression of ChR2, localized to PVs or SOMs (*Figure 6—figure supplement 1B*, c PV-Cre; N = 183 neurons in 3 mice, specificity = 67 ± 1%, efficiency = 76 ± 5%. SOM-Cre: N = 202 neurons in 4 mice, specificity = 90 ± 3%, efficiency = 81 ± 4%). Neurons were identified as PVs or SOMs if they responded to brief (5 ms) flashes of light with spikes within 1.5–4.5 ms of laser pulse onset (*Figure 6A,D*).

Both PVs and SOMs exhibited SSA, evidenced by a significant reduction in standard tone-evoked FR compared to the deviant tone response (*Figure 6B,C,E,F*, PV: N = 16, Δ = −32%, p2 = 0.023, z = −2.5, C = 2. SOM: N = 28, Δ = −41%, p2 = 0.002, z = −3.3, C = 2. Signed-rank test). The SSA index was not significantly different between PVs and SOMs (*Figure 6G*, neurons responsive to both tones A and B—PV: N = 5, SOM: N = 12. PV and SOM: p2 > 0.05, C = 2. Rank sum test) and both were similar to the mean SSA index in putative excitatory neurons (*Figure 6G*—Exc: N = 67. Exc vs PV: p > 0.05, z = 0.7, C = 2. Exc vs SOM: p > 0.05, z = 0.4, C = 2). PVs and SOMs exhibited some differences in relative response changes between the deviant, the standard, and the equal tones (*Figure 6*, *Figure 6—figure supplement 2B,D*); PVs' response to the equal tones did not decrease significantly as compared to deviant tones (N = 16 p2 > 0.05, z = −1.7, C = 2), whereas SOMs adapted in their response to equal tones (Δ = −36%, p2 = 0.049, z = −2.3, C = 2), and then further to standard tones (N = 28, Δ = −49%, p2 = 0.022, z = −2.6, C = 2). These results suggest that SOMs may adapt at a faster time scale than PVs with repeated presentation of tones.

## Adapting inhibitory interneurons facilitate SSA in excitatory neurons in a cortical network model

Our results of recordings from PVs and SOMs present a surprising finding that PVs and SOMs adapt in response to repeated tones, countering our initial hypothesis that SOMs saturate in responses to the deviant, or facilitate with repeated presentation of a tone. How can an adapting interneuron contribute to added adaptation in excitatory neurons? To address this question, we next developed a model of coupled excitatory–inhibitory neuronal populations. Excitatory and inhibitory neurons form tight mutually coupled networks in A1, and we hypothesized that through differential post-synaptic integration by excitatory neurons, interneurons can amplify adaptation in excitatory neurons.

As a proof-of-principle that would account for our findings that PVs and SOMs exhibit similar magnitude of SSA, yet have a differential effect on SSA in putative excitatory neurons, we constructed a simplified model of mutually coupled inhibitory–excitatory neuronal populations. We tested how responses of the model putative excitatory neurons are affected by manipulation of activity of PVs or SOMs (*Figure 7A*). Thalamocortical tone-evoked inputs were modeled including an adaptation term and resulted in reduced responses of excitatory, PV, and SOM populations to repeated tones (*Figure 7—figure supplement 1A,B*). The model replicated the differential effects of manipulation of PV and SOM activity on responses to standard and deviant tones in putative excitatory neurons (*Figure 7B–E*): when PVs were suppressed optogenetically, the responses to both the standard and the deviant tones increased (*Figure 7B,C*). By contrast, when SOMs were suppressed, although the spontaneous FR and standard tone-evoked FR were elevated, the responses to the deviant tone remained constant, whereas the responses to the standard tone increased (*Figure 7D,E*). SOMs have been shown to inhibit PVs (*Cottam et al., 2013*; *Pfeffer et al., 2013*; *Sturgill and Isaacson, 2015*). Including inhibition between SOMs and PVs did not affect the model outcome, with suppression of PVs resulting in suppression of excitatory responses to both the standard and the deviant, and suppression of SOMs driving specific suppression of excitatory responses to the standard, but not the deviant (*Figure 7—figure supplement 2*).

An explanation for the difference of the effects of PVs and SOMs can be provided by examining the combined transfer function between pre-synaptic inputs and post-synaptic activity of excitatory

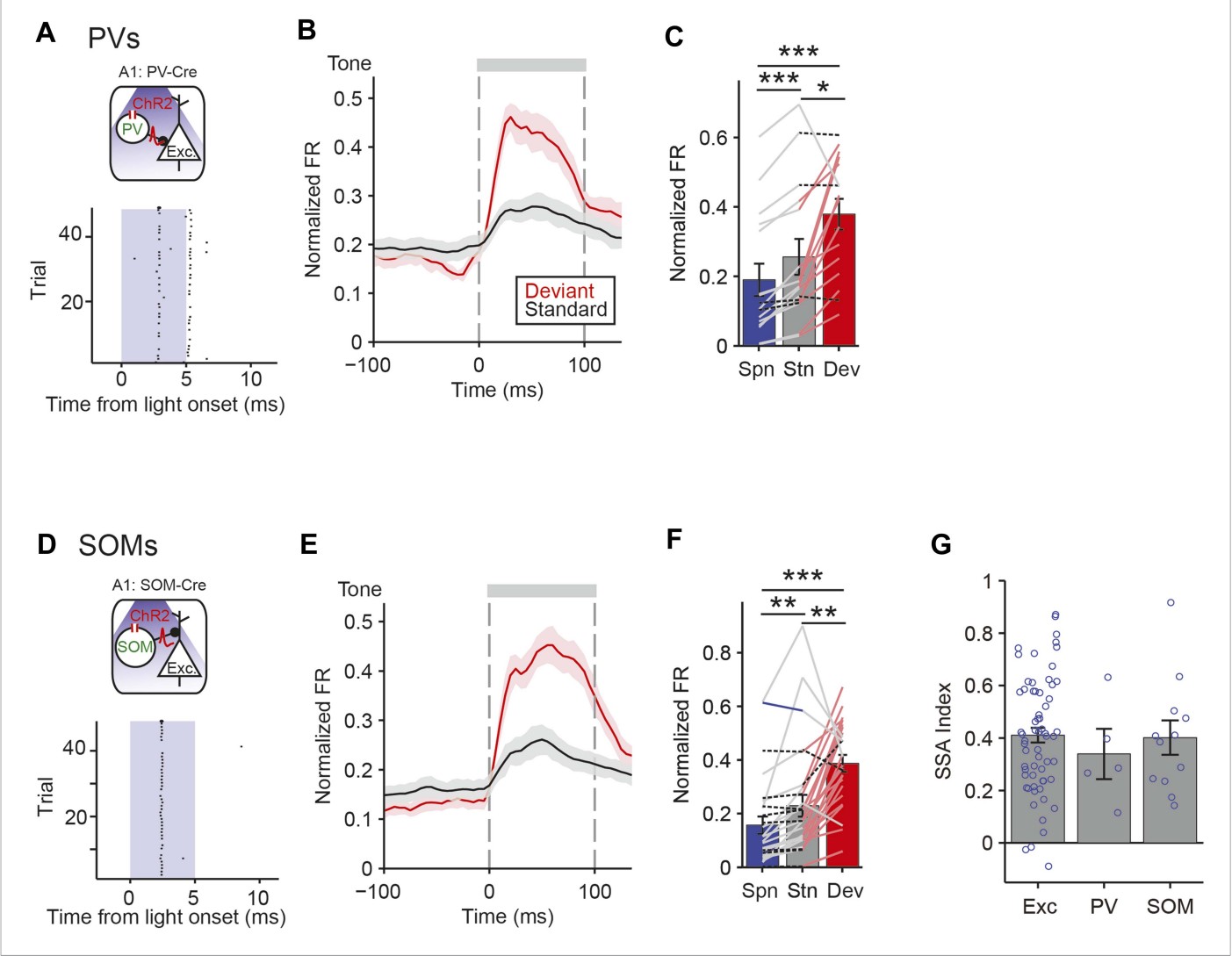

**Figure 6**. PV and SOM interneurons exhibit SSA. (**A**, **D**) Optogenetic methods. A1 was injected with AAV-FLEX-ChR2-tdTomato. During experiments, an optic fiber was positioned to target A1 and neuronal activity was recorded using a multichannel silicon probe in A1. Top diagram: blue light (473 nm) excites PVs in PV-Cre mice or SOMs in SOM-Cre mice. Bottom: peri-stimulus spike raster of a representative optogenetically identified PV (top) or SOM (bottom). Shaded region, blue light on. (**A**) PV-Cre. (**D**) SOM-Cre. (**B**, **E**) PSTH of PVs (**B**) or SOMs (**E**) FR response to deviant (red) and standard (black) tones. Normalization and dashed lines as in *Figure 4A,B*. (**C**, **F**) Mean PVs (**C**) or SOMs (**F**) FR response over the 100 ms of deviant (red) and standard tones (gray), and 100 ms of spontaneous activity prior to tone onset (blue). Each line represents a single neuron's response to each conditions, and its color indicates the magnitude of significant differences between two conditions; pink, gray, blue, and dashed black lines indicate a greater response to deviant tone, standard tone, silence and no significant change, respectively. (**G**) Mean SSA index of putative excitatory neurons, PVs, and SOMs. Circles represent SSA index values of individual neurons.

The following figure supplements are available for figure 6:

**Figure supplement 1**. Optical tagging of PVs and SOMs.

**Figure supplement 2**. PVs and SOMs have different adaptation profiles for equal probability tones.

neurons separately for PVs and SOM suppression (*Figure 7A*, insets): light-driven modulation of PV activity has the same effect on excitatory neuron responses at spontaneous, standard tone-evoked, and deviant tone-evoked activity (*Figure 7A*, left inset). Spontaneous, standard, and deviant input levels all fall within the linear portion of the transfer function between inputs and change in the excitatory neuron activity. On the other hand, for SOMs, modulation of their activity in the deviant

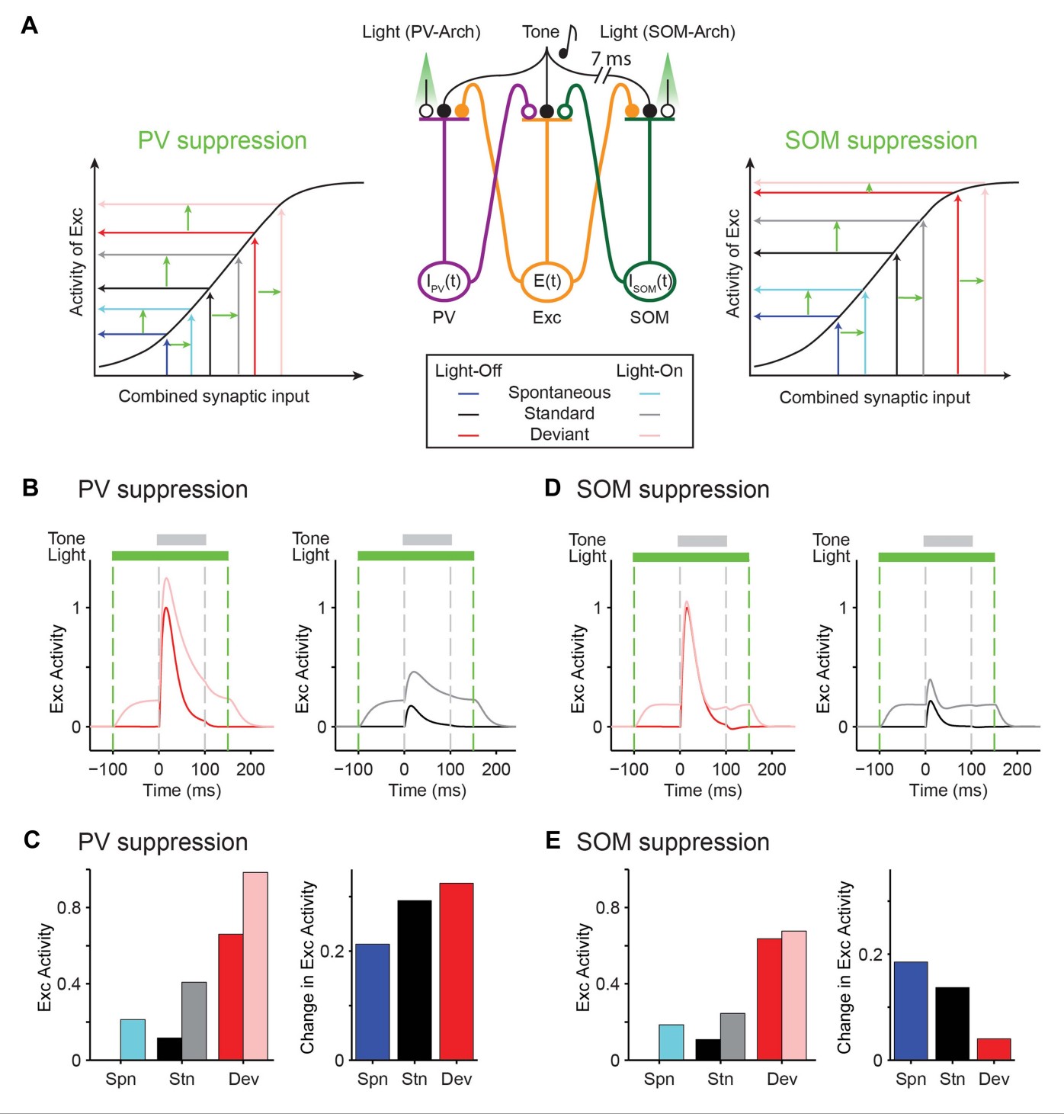

**Figure 7**. Mutually coupled excitatory-PV-SOM neuronal model accounts for differential effects of PVs and SOMs on SSA in putative excitatory neurons. (**A**) Center: diagram of coupled network model. Excitatory (Exc) and two types of inhibitory interneurons (PV and SOM) receive tone-evoked inputs. They make reciprocal connections on each other; Exc makes excitatory synapses on PV or SOM; PV and SOM inhibit Exc. Closed circles: excitatory synapses. Open circles: inhibitory synapses. Orange outlines: excitatory input–output pathway. Purple outlines: PV input–output pathway. Green outlines: SOM input–output pathway. The effect of optogenetic modulation was modeled as an additional input current delivered to inhibitory neuronal populations. Adaptation was modeled as decaying synaptic coefficient with slow adaptation. Left and right inset plots: combined input–output transfer function that represents the transformation between synaptic inputs and the activity of excitatory neurons. The values of inputs are depicted by arrows for the spontaneous and tone-evoked activity in response to deviant and standard tones under light-off (dark color) and light-on (light color) conditions, with

*Figure 7. continued on next page*

*Figure 7. Continued*

change due to light highlighted by light green arrows. (**B**, **D**) Tone-evoked responses of model neuronal excitatory population to deviant (red) and standard tones (gray), that is, the first and fourth consecutive tone presented, under light-off (dark colors) and light-on (light colors) conditions. Dashed lines indicate light onset and offset (green) and tone onset and offset (gray). (**B**) Light suppresses PVs. (**D**) Light suppresses SOMs. (**C**, **E**) Left: spontaneous FR (blue) and standard (black) and deviant (red) tone-evoked FRs on light-off (dark colors) and light-on (light colors) conditions. Right: mean difference between responses on light-on and light-off conditions. (**C**) Light suppresses PVs. (**E**) Light suppresses SOMs.

The following figure supplements are available for figure 7:

**Figure supplement 1**. Adaptation to repeated tones in model excitatory and inhibitory neurons.

**Figure supplement 2**. Excitatory–inhibitory model with inhibitory inputs from SOM to PV population accounts for differential effects of PVs and SOMs on SSA in putative excitatory neurons.

tone-evoked regime drives small to no changes in excitatory neuronal activity, whereas modulation of SOM activity in the spontaneous and standard tone-evoked regime drives significant changes in excitatory neuronal activity (*Figure 7A*, right inset). The deviant tone-evoked activity falls on the saturating part of the input–output transfer function, whereas the standard tone-evoked and spontaneous inputs fall on the linear part of the transfer function. Then, shifts in SOM inputs due to photosuppression evoke small changes during deviant tone responses, but larger changes during either standard or spontaneous activity. Either PV or SOM manipulation would result in reduction of combined SSA of excitatory neurons.

## Discussion

The majority of neurons in the auditory cortex selectively reduce their responses to frequent, but not rare sounds, exhibiting SSA. However, the cortical mechanisms involved in the production and stimulus-specificity of SSA within the auditory cortex are not well understood. Here, we found that, in addition to adaptation at the level of thalamocortical inputs, two distinct types of interneurons, PVs and SOMs, differentially contributed to SSA in the primary auditory cortex. Optogenetic suppression of either PVs or SOMs led to a reduction in SSA in putative excitatory neurons (*Figure 3*). Suppression of PVs led to an equal increase in the FR of the putative excitatory neurons in response to the standard tone and the deviant tone (*Figure 4*). By contrast, suppression of SOMs significantly increased the response to the standard tone but lacked a significant effect on the response to the deviant tone (*Figure 4*). This series of findings expands on the 'adaptation in narrowly tuned units' model, which proposes that repeated presentation of the standard stimulus drives adaptation within more narrowly tuned inputs, such as thalamocortical inputs (*Mill et al., 2011*; *Taaseh et al., 2011*; *Nelken, 2014*). Our data indicate dual effects of cortical inhibition on SSA: (1) PVs contribute to SSA by providing a constant level of inhibition, resulting in a *relatively* higher inhibitory drive during the presentation of the standard, as compared to the deviant. Taking into account the non-linear synaptic input to FR output function of a typical pyramidal neuron, the constant inhibition amplifies the effect of thalamocortical depression in suppressing the response of the neuron to repeated stimulus (*Figure 7A*). (2) The selective increase of the inhibitory drive from SOMs for standard stimulus as compared to the deviant stimulus responses might be explained by a shift in the non-linear transfer function between inputs to SOMs and their outputs to excitatory neurons, possibly due to facilitation of SOM-to-excitatory neuron synapses (*Beierlein et al., 2003*; *Silberberg and Markram, 2007*) (*Figure 7A*).

Surprisingly, we found that, despite the differential effect of PV and SOM suppression on tone-evoked responses in putative excitatory neurons, both PVs and SOMs exhibit SSA. This finding is consistent with previous results that found that thalamocortical synapses onto inhibitory neurons and synapses from inhibitory neurons to excitatory cells can be depressing (*Tan et al., 2008*; *Ma et al., 2012*). How does suppression of these interneurons result in differential reduction in SSA in excitatory neurons? Our model provides an intuition for this effect: the mutually coupled excitatory–inhibitory network model demonstrates that the observed differential effects of PV and SOM suppression may be due to their differential action on excitatory neuronal responses in the

unadapted and adapted state (*Figure 7*). Tone-evoked responses of PVs would fall on the linear portion of the transfer function between PV activity and excitatory neuron depolarization, while the same tones maximally affect inputs from SOMs onto excitatory neurons, with SSA shifting the inputs to the linear, more sensitive range of inputs from SOMs. Thus, SSA may serve an additional function: to adjust the responses of neurons in a range that is more sensitive to small changes in the inputs from both excitatory and inhibitory neuronal populations. More generally, the simulation demonstrates that a circuit element, such as PVs or SOMs, that itself adapts may further amplify adaptation in the excitatory neurons.

To estimate the differential contribution of PVs or SOMs inputs to the excitatory neurons, we measured the difference in the FR of neurons due to optogenetic partial suppression of their firing. This measurement provides an estimate of the change in the FR of the putative excitatory neurons with the change in combined inputs to the inhibitory neurons, thereby allowing estimation of the synaptic transfer function (*Figure 7A*, insets). A simple biologically plausible network incorporating these transfer functions can reproduce the observed responses (*Figure 7*). There are several caveats to this interpretation. First, the FR may not linearly translate onto synaptic input strength because of the spiking non-linear rectification between the inputs and outputs of the putative excitatory neuron: a small change in FR in the low-FR regime might correspond to a greater change in the synaptic drive than a similar-sized change in FR in the high-FR regime. However, our findings would still hold were this the case: in examining the effect of SOM suppression on response to the deviant, the actual difference in the synaptic drive between the deviant and the standard would then be even greater than observed. At the other end of the non-linearity, the analysis of neuronal responses sorted based on their FR to the standard tone and the deviant tone revealed that the 'ceiling effect' would not contribute to a decreased effect of photostimulation on the response to the deviant in SOM-Cre mice (*Figure 4—figure supplements 4–7*). Second, PVs and SOMs may inhibit not only the excitatory neurons, but also each other. SOMs make synapses onto PVs (*Isaacson and Scanziani, 2011*; *Ma et al., 2012*; *Cottam et al., 2013*), thereby potentially suppressing them with repeated presentation of the standard. Therefore, when SOMs are suppressed, some PVs may be disinhibited and provide a stronger suppression of excitatory neurons. The null effect on responses to the deviant tone during SOM suppression could result from a combination of increase in inhibition from disinhibited PVs in addition to reduced inhibition of SOMs onto excitatory neurons. Including inhibition from SOMs to PVs in the proof-of-principle model supported experimental findings (*Figure 7—figure supplement 2*). Third, other interneuron types, such as vasopressin-positive interneuron may be involved in the circuit (*Pi et al., 2013*), and the changes that we observe may reflect several inhibitory stages of processing.

One must be cautious in translating the data from our experiments as a strict description of neuronal activity in awake animals, as our results were based on recordings from mice under light isoflurane anesthesia. Other forms of anesthesia, such as pentobarbital-based (*Cheung et al., 2001*; *Gaese and Ostwald, 2001*), ketamine (*Otazu et al., 2009*) and high concentrations of isoflurane (*Cheung et al., 2001*; *Ter-Mikaelian et al., 2007*), can affect multiple aspects of sound-evoked responses in the auditory cortex. Nonetheless, our results are likely to extend for awake mice, since isoflurane anesthesia-induced effects on neuronal activity decrease as the concentration of isoflurane is reduced to the levels used in our recordings (*Land et al., 2012*). In addition, all recordings and manipulations were performed under identical anesthetic conditions, and our conclusions are based on the relative comparison of the effects of suppressing PVs and SOMs, which are expected to hold under awake conditions (*Centanni et al., 2013*).

While not demonstrated directly, SSA has been linked to detection of deviant sounds (*Ulanovsky et al., 2003*), which may be facilitated by a relatively enhanced neuronal response to a change in the ongoing sound (*Nelken et al., 2003*; *Winkler et al., 2009*; *Grimm and Escera, 2012*). By suppressing the responses to a frequently presented tone, the responses of neurons to a rare stimulus become relatively enhanced. However, whether and how modulating SSA in the auditory cortex affects auditory behavior has not yet been tested. Inhibitory interneurons may prove to have a complementary role in shaping auditory perception in addition to receptive field reorganization driven by synaptic plasticity (*Froemke et al., 2013*). The use of optogenetic methods to test the function of inhibitory interneurons in SSA overcomes the limitations of lesion or pharmacological studies (*Elliott and Trahiotis, 1972*; *Duque et al., 2014*), which only allow for prolonged, non-selective inactivation (*Moore et al., 2001*). By combining optogenetic manipulation of interneuron activity with behavioral measurements, future experiments will explore whether interneuron-mediated

SSA indeed affects the auditory behavior of the subject, such as enhanced ability to detect unexpected events.

## Materials and methods

### In vivo experimental preparation

#### Animals

All experiments were performed in adult male mice (Jackson Laboratories, Bar Harbor, ME, United States; age, 12–15 weeks; weight, 22–32 g; PV-Cre mice, strain: *B6;129P2-Pvalbtm1(cre)Arbr/J*; SOM-Cre: *Ssttm2.1(cre)Zjh/J*) housed at 28°C on a 12-hr light:dark cycle with water and food provided ad libitum. In PV-Cre mice, Cre recombinase (Cre) is expressed in parvalbumin-positive interneurons; in SOM-Cre mice, Cre is expressed in somatostatin-positive interneurons (*Taniguchi et al., 2011*). This study was performed in strict accordance with the recommendations in the Guide for the Care and Use of Laboratory Animals of the National Institutes of Health. All of the animals were handled according to a protocol approved by the Institutional Animal Care and Use Committee of the University of Pennsylvania (Protocol Number: 803266). All surgery was performed under isoflurane anesthesia, and every effort was made to minimize suffering.

#### Viral vectors

Modified AAVs were obtained from Penn Vector Core (Philadelphia, PA, United States). Modified AAV encoding Arch under FLEX promoter was used for selective suppression of PVs or SOMs (catalog number AV-9-PV2432, AAV9.CBA.Flex.Arch-GFP.WPRE.SV40, Addgene22222, serotype 2/9) (*Chow et al., 2010*). Modified AAV encoding GFP alone under FLEX promoter was used as a control for the specific action of Arch on the neuronal populations (catalog number AV-9-ALL854, AAV9.CAG.Flex. eGFP.WPRE.bGH, Allen Institute 854, serotype 2/9). Modified AAV encoding ChR2 under FLEX promoter was used for selective excitation of PVs or SOMs (catalog number AV-9-18917P, AAV9. CAGGS.Flex.ChR2-tdTomato.WPRE.SV40, Addgene18917, serotype 2/9).

#### Virus injection

2–3 weeks prior to the start of experimental recordings, a 0.5-mm diameter craniotomy was drilled over primary auditory cortex (2.6 mm caudal and 4.1 mm lateral from bregma) under aseptic conditions while the mouse was anesthetized with isoflurane. A 750 nl bolus of AAV in water was injected into A1 (1 mm ventral from pia mater) using a stereotaxic syringe pump (Pump 11 Elite Nanomite, Havard Apparatus, Holliston, MA, United States). The craniotomy was covered with bone wax and a small custom head-post was secured to the skull with dental acrylic.

#### Electrophysiological recordings

All recordings were carried out inside a double-walled acoustic isolation booth (Industrial Acoustics, Bronx, NY, United States). Electrodes were targeted to A1 on the basis of stereotaxic coordinates and in relation to blood vessels. In electrophysiological recordings, the location was confirmed by examining the click and tone-pip responses of the recorded units for characteristic responses of neurons in core auditory areas, as described previously by our group in the rat (*Carruthers et al., 2013*) and by other groups in the mouse (*Linden and Schreiner, 2003*; *Guo et al., 2012*; *Marlin et al., 2015*). While the electrodes were targeted to A1, some recordings may include data from the anterior auditory field, adjacent to A1 (*Linden et al., 2003*). Mice were placed in the recording chamber, anesthetized with isoflurane, and the headpost secured to a custom base, immobilizing the head. After drilling a craniotomy and creating a durotomy exposing auditory cortex, a silicon multi-channel probe (A1x32-Poly2-5mm-50s-177 [Poly-2] or A1x32-tri-5mm-91-121-A32 [Triode], Neuronexus Ann Arbor, MI, United States) was slowly lowered to between 750 µm and 1 mm into the cortex, perpendicular to the cortical surface and used to record electrical activity. Raw signals from 32 channels were bandpass filtered at 600–6000 Hz and thresholded for spike analysis, or at 10–300 Hz for LFP and CSD analysis (Poly-2 probe only), digitized at 32 kHz and stored for offline analysis (Neuralynx, Bozeman, MT, United States). Common-mode noise was removed by referencing a probe inserted in the brain outside the auditory cortex. On the Poly-2 probe, two rows of 16 electrodes each on a single shank were arranged such that each electrode site was 50 µm away from all three closest neighbors. This arrangement allowed us to record densely across depth, that is, one electrode for every 25 µm in

depth. On the triode, electrodes were arranged in groups of three equidistant sites, forming an equilateral triangle (25-μm separation). The triodes were separated vertically by 91-μm center-to-center distance, spanning 1 mm, with two additional single sites, one on each end.

## Unit identification

Spike sorting was performed using commercial software (Offline Sorter, Plexon, Dallas, TX, United States) (*Carruthers et al., 2013*). In order to improve isolation of single units from recordings using low-impedance probes, spiking activity was sorted across three (triode, 25-μm separation) or four (poly-2, 50-μm separation) adjacent electrode sites (*Niell and Stryker, 2008*; *Olsen et al., 2012*). We used a stringent set of criteria to isolate single units from multiunit clusters (*Otazu et al., 2009*; *Bizley et al., 2010*; *Brasselet et al., 2012*; *Durand et al., 2012*; *Carruthers et al., 2013*; *Picard et al., 2014*; *Carruthers et al., 2015*). Single-unit clusters contained <1% of spikes within a 1.0-ms interspike interval, and the spike waveforms across 3 or 4 channels had to form a visually identifiable distinct cluster in a projection onto a three-dimensional subspace. Putative excitatory neurons were identified based on their expected response patterns to sounds and the lack of significant suppression of the spontaneous FR due to light (*Lima et al., 2009*; *Moore and Wehr, 2013*). While this subpopulation may still contain inhibitory neurons, only 2% of all recorded neurons were significantly photosuppressed at baseline (one-sided paired t-test, significance taken at $p < 0.05$). The low impedance of the extracellular probes precluded us from conducting a more detailed analysis of cortical subpopulations based on the spike waveform (*Bartho et al., 2004*; *Moore and Wehr, 2013*).

## Acoustic stimulus

Stimuli were delivered via a magnetic speaker (Tucker-David Technologies, Alachua, FL, United States), directed toward the mouse's head. Speakers were calibrated prior to the experiments to ±3 dB over frequencies between 1 and 40 kHz, by placing a microphone (Brüel and Kjaer, Denmark) in the location of the ear contralateral to the recorded A1 hemisphere, recording speaker output and filtering stimuli to compensate for acoustic aberrations (*Carruthers et al., 2013*). First, to measure tuning, a train of 50 pure tones of frequencies spaced logarithmically between 1 and 80 kHz, at 65-dB sound pressure level (SPL) relative to 20 μPa, in pseudorandom order, was presented 20 times. Each tone was 100-ms long, with an inter-stimulus interval (ISI) of 300 ms. Frequency response functions were calculated online for several multiunits, and two frequencies (separated by 0.39 octaves), which elicited spiking responses of similar strength, were selected as tone A and B. Next, a series of stimuli composed of tones A and B were presented in interleaved blocks, repeated four times. Each oddball stimulus consisted of a train of 653 A and B tones (100-ms long, 300-ms ISI, 65-dB SPL). In oddball stimulus 1, 90% of the tones were A (standard), while 10% of the tones were B (deviant). We used a frozen sequence of standard and deviant tones in pseudorandom order and counterbalanced with respect to the number of standard tones preceding each deviant. In oddball 2, the probabilities of tones A and B were reversed so that tone B was the standard and A the deviant. In the equal probability stimulus, A and B each comprised 50% of tones.

## Light presentation

An optic fiber was use to direct 532-nm laser light (Shanghai Laser & Optics Century, China). After positioning the silicon probe, an optic fiber was placed over the surface of auditory cortex. To limit Becquerel effect artifacts due to light-striking electrodes, we positioned the optical fiber parallel to the silicon probe (*Han et al., 2009*; *Kvitsiani et al., 2013*). During every fifth tone of the oddball and equal probability stimuli, light was cast over A1 to suppress interneurons. The light onset was 100 ms prior to tone onset, and lasted for 250 ms. At 180 mW/mm², light pulses were intense enough to significantly modulate multiunit activity throughout all cortical layers. The effect of optical stimulation was not significant for responses to subsequent tones (*Figure 3—figure supplement 1*).

## Immunohistochemistry

Brains were post-fixed in paraformaldehyde (4%, PFA) and cryoprotected in 30% sucrose. Coronal sections (40 μm) were cut using a cryostat (CM1860, Leica, Allendale, NJ, United States), washed in PBS containing 0.1% Triton X-100 (PBST; three washes, 5 min), incubated at room temperature in blocking solution (for PV, 10% normal goat serum and 5% bovine serum albumin in PBST; for SOM, 10% normal goat serum with 0.1% sodium azide and 2% cold water fish gelatin in PBS; 3 hr), and then incubated in primary antibody diluted in blocking solution overnight at 4°C. The following primary antibodies were used: anti-PV (PV 25 rabbit polyclonal, 1:500, Swant, Switzerland) or anti-SOM

(AB5494 rabbit polyclonal, 1:200, Millipore, Billerica, MA, United States). After incubation, sections were washed in blocking solution (three washes, 5 min), incubated for 2 hr at room temperature with secondary antibodies (Alexa 594 goat anti-rabbit IgG; for PV 1:1000 and SOM 1:400), and then washed in PBS (three washes, 5 min each). Sections were mounted using Fluoromount-G (Southern Biotech, Birmingham, AL, United States) and confocal images were acquired (Leica SP5). Cells were identified in independent fluorescent channels and subsequently scored for co-localization by hand using ImageJ's cell counter plug-in. Transfection efficiency is the percent of antibody-labeled neurons, which are co-labeled with GFP. Transfection specificity is the percent of GFP-expressing neurons, which are co-labeled with the antibody.

## In vitro experimental preparation

### Slice preparation
Acute brain slices were prepared from mice using standard techniques essentially as previously described (*Goldberg et al., 2011*). Mice were anesthetized via inhaled isoflurane and then transcardially perfused with 10 ml of oxygenated, ice-cold artificial cerebrospinal fluid (ACSF) at a rate of 5 ml/min, that contained, in mM: 87 NaCl, 75 sucrose, 2.5 KCl, 1.25 $NaH_2PO_4$, 26 $NaHCO_3$, 10 glucose, 0.5 $CaCl_2$, 4 $MgSO_4$. Slices (300-μm thick) were cut on a Leica VT1200S and incubated in cutting solution in a holding chamber at 32°C for approximately 30 min followed by continued incubation at room temperature prior to electrophysiological recording, at which point slices were transferred to a submersion-type recording chamber attached to the microscope stage. ACSF used for recording contained, in mM: 125 NaCl, 2.5 KCl, 1.25 $NaH_2PO_4$, 26 $NaHCO_3$, 10 glucose, 2 $CaCl_2$, and 1 $MgSO_4$. The solution was continuously bubbled with 95% $O_2$ and 5% $CO_2$ throughout cutting, slice incubation, and recording, so as to maintain a pH of approximately 7.4.

### Electrophysiology
Cells were identified via GFP expression under epifluorescence microscopy and subsequently visualized using a 40×, 0.8 NA water-immersion objective (Olympus, Center Valley, PA, United States) on an Olympus BX-61 upright microscope equipped with infrared differential interference contrast optics. Recordings were performed using the whole-cell patch clamp technique. Access resistance (Ra) was <25 MΩ upon break-in; data obtained from a given cell were rejected if Ra changed by >20% during the course of the experiment. Internal solution contained, in mM: potassium gluconate, 130; potassium chloride, 6.3; EGTA, 0.5; $MgCl_2$, 1.0; HEPES, 10; Mg-ATP, 4; Na-GTP, 0.3; biocytin, 0.1%. Osmolarity was adjusted to 285–290 mOsm using 30% sucrose. Voltage was recorded using a MultiClamp 700B amplifier (Molecular Devices, Union City, CA, United States), lowpass filtered at 10 kHz, digitized at 16-bit resolution (Digidata 1550, Axon Instruments, Sunnyvale, CA, United States), and sampled at 20 kHz. pCLAMP 10 software was used for data acquisition, and analysis was performed using the Clampfit module of pCLAMP.

### Optogenetics
Cells were illuminated with a 561-nm solid state laser (Coherent, Santa Clara, CA, United States) routed to the standard X-Y galvanometer of a two-photon microscope (Bruker Corporation, Billerica, MA, United States) via a single-mode fiber. Illuminance at the specimen was estimated using a 10-μm pinhole aperture (Edmund Optics, Barrington, NJ, United States) and a photodiode power sensor (Thorlabs, Newton, NJ, United States).

## In vivo neuronal response analysis

### Tone response FR
For each putative excitatory neuron, the spontaneous FR and tone-evoked FRs were measured as the mean FR over 50 ms pre- and post-tone onset, respectively. For each identified interneuron, FRs were measured 100 ms pre- and post-tone onset. FR was measured separately for each tone, A and B, as standard, deviant, and equal probabilities, and for light-off and light-on trials. FR normalization was carried out separately for each tone, A and B, for each neuron by dividing the response under all conditions by the maximum FR (across 5-ms bins) of the deviant tone, light-off condition. Performing this normalization by dividing response in all conditions by the mean, rather than maximum FR of the deviant tone, light-off condition did not alter significant results (*Figure 4—figure supplement 8*). For all FR analyses, each neuron's responses to tones A and B were treated separately, and each was only

included if the light-off deviant tone-evoked FR was significantly greater than the spontaneous FR (Wilcoxon signed-rank test p < 0.05). Further, tone responses were only included in analysis if the neuronal FR during each oddball stimulus exceeded 0.02 Hz, and the neuron was significantly tuned to the tone. Tuning was considered significant if the spike count in response to a tone (A or B) was significantly higher than the pool of spike counts across all tones outside one octave band centered on tones A and B (N = 42, t-test, p1 < 0.05). Population responses in each condition were measured as the mean and standard error of FRs across tone responses in each experimental group.

## SSA index

For each neuron, SSA index is a measure of the strength of SSA based on its mean FR with respect to tone probability. FRs to tones A and B were summed according to their standard or deviant probability within each oddball stimulus (*Ulanovsky et al., 2003*). Thus, SSA index was computed as:

$$SSA\ \ Index = \frac{(D_A + D_B) - (S_A + S_B)}{D_A + D_B + S_A + S_B},$$

where $S$ and $D$ indicate the mean FR for standard and deviant trials, respectively, and their subscripts indicate the tone frequency condition. SSA index was computed separately for light-off and light-on conditions. Population SSA indices were measured as the mean and standard error of SSA indices across all neurons of each population. Criteria for inclusion in the analysis were the same as in *tone response FR* analysis described above, with the added criterion that the deviant tone-evoked FR must be greater than spontaneous FR for each of tones A and B (Wilcoxon signed-rank test p < 0.05).

## Localization of cortical layers and CSD

To calculate the CSD, the net current density moving through cortical tissue at 32 positions along the cortical axis was calculated based on LFPs of responses to tones recorded on each electrode, by using the second order central finite difference to calculate the second spatial derivative across the LFPs over the vertically arranged electrodes (*Szymanski et al., 2009*). Across the CSD profile, the deepest current sink corresponds to the thalamo-recipient granular layer (*Kaur et al., 2005*; *Szymanski et al., 2009*) allowing us to reconstruct the laminar location of recorded neurons. Neurons recorded on electrodes falling within the deepest sink were assigned to the granular layer, while those superior and inferior were assigned to the supra-granular and infra-granular layers (*Figure 1E,F*, *Figure 4—figure supplement 3*). The tone-evoked amplitude of the CSD was measured by first calculating root mean square of each channel during the first 50-ms post-tone onset, and then calculating the mean across all electrodes determined to fall within either the deepest short latency sink (granular layer) or pooled across all electrodes either above (supra-granular layer) or below (infra-granular layer). For each session, the granular layer CSD amplitude for all tone conditions was normalized across conditions by the deviant tone, light-off condition, and the mean across sessions was statistically analyzed. The SSA index was calculated as described in *SSA index* on the basis of the amplitude.

## Statistical tests

For all statistical tests in which N ≥ 30, we applied the Student's *t*-test (Matlab, Mathworks, Natick, MA, United States) unless specified otherwise, and reported the p-value, degrees of freedom, and t-statistic. For all tests with N < 30, sample variance was tested for normality using the Komogorov–Smirnov test. If any group's variance was non-normal, we applied a non-parametric test, for example, Wilcoxon sign rank or rank sum test (Matlab), and provided the z-statistic for any group with a normal distribution. For all tests, Bonferroni correction was applied for multiple comparisons and reported as 'C = X' where X is the factor by which the p-value was adjusted. Statistical tests were single-tailed if there was a reasonable prior expectation about the direction of the difference between samples. p1 refers to one-sided, and p2 refers to two-sided statistics set. In all figures, single, double, and triple stars indicate p < 0.05, 0.01, and 0.001, respectively. Error bars in all figures represent the standard error of the mean, unless otherwise noted.

## Excitatory–inhibitory network model

We constructed models of the excitatory–inhibitory neuronal circuit to understand the coupling of excitatory interneurons with PV and SOM interneurons. We constructed firing-rate models based on Wilson–Cowan dynamics (*Staiger et al., 1996*; *Xu et al., 2010*, *Rudy et al., 2011*). The parameters

were chosen in order to achieve a match to experimental data. The mean activity level of each population was modeled as:

$$\frac{dE}{dt} = \frac{1}{\tau_E} \left[ -E(t) + (k-r)S(j_{ETone}(t) + S_{inh}(j_{IE}I(t))) \right],$$

$$\frac{dI}{dt} = \frac{1}{\tau_I} \left[ -I(t) + (k-r)S(j_{inh}(t) + j_{ITone}(t) + j_{EI}E(t)) \right],$$

where $E(t)$ is the activity of the excitatory population; $I(t)$ is the activity of the inhibitory population; $S(x)$ is the transfer function between the combined 'synaptic' input and the neuronal FR. $S(x)$ is linear with respect to intermediate inputs, but imposes a minimum and maximum activation limits. $S_{inh}(x)$ is the transfer function between the inhibitory FR and excitatory post-synaptic current; $j_{EI}$ and $j_{IE}$ are excitatory–inhibitory and inhibitory–excitatory synaptic weights (0.2 and −1.0 for PVs, 0.05 and −0.39 for SOMs, respectively); $j_{ETone}(t)$ and $j_{ITone}(t)$ are tone-evoked input currents to excitatory and inhibitory neurons, respectively, modeled as 50-ms long exponentially decaying inputs of maximum amplitude 3 (delayed by 7 ms for SOMs, which do not receive direct thalamic inputs, relative to PVs, which receive direct thalamic inputs); $\tau_E$ (10 ms) and $\tau_I$ (10 ms) are synaptic time constants for excitatory and inhibitory neurons; k and r represent the maximum and minimum FR of neurons, respectively, (k = 15, r = 1); $j_{inh}(t)$ is the negative input to inhibitory neurons due to Arch. The optogenetic modulation was modeled as a unitary 250-ms pulse. To capture the differences in inputs due to repeated tone exposures, we modeled thalamic inputs reflecting the tone inputs with synaptic depression. We modeled the conductance of the thalamic projections, $g_{Inp}$, as changing according to the equation:

$$\frac{dg_{Inp}}{dt} = \left( g_0 - g_{Inp} \right) \Big/ T_g - \left( g_{Inp}r \right) \Big/ T_r,$$

where $g_0$ is the maximum conductance ($g_0 = 1$), r is the gating coefficient representing tone-evoked thalamic input, $T_g$ is the time scale for replenishment ($T_g$ = 3 s), $T_r$ is the time scale for depletion ($T_r$ = 80 ms). We took r to be a step function with an exponential decay (with 40-ms time constant and amplitude of 3). The full input to auditory cortical neural populations is then equal to $g_{Inp}r$. In a train of four tones, the first tone-evoked response was taken as the deviant tone, and the fourth tone as the standard tone.

For the inhibitory-to-excitatory inputs, we used a sigmoidal transfer function and showed the existence of parameter regimes consistent with our results. For PVs, we used a sigmoid of the form:

$$S_{PV}(r_{PV}) = \frac{1}{1 + \exp[-p(r_{PV} - \theta)]},$$

where p = 0.3 and θ = 9. This gives a facilitating response at low input levels and a linear response at high input levels. For SOMs, we used a hyperbolic tangent that provided a saturating non-linearity:

$$S_{SOM}(r_{SOM}) = \frac{1 - \exp[-2r_{SOM}/s]}{1 + \exp[-2r_{SOM}/s]},$$

where s = 3. For visualization, the baseline FR of neurons was removed and the peak response to a 'deviant' tone without optogenetic manipulation normalized to 1.

We also constructed a model with additional coupling between the PV and SOM interneurons using a generalization of the above dynamics, which may be written as:

$$\frac{dN_i}{dt} = \frac{1}{T_i} \left( -N_i + (k-r)S\left( j_{tone,i}(t) + j_{ext,i}(t) + \sum_k j_{ki} * S_k(N_k) \right) \right),$$

where $N_i$ is the FR of the ith population (EXC, PV, SOM), $T_i$ = 10 ms is the time constant for each population, k = 15, r = 1, S has different maximum and minimum values for each population ($x_{min,E}$ = −1, $x_{max,E}$ = 1.75, $x_{min,PV}$ = −0.5, $x_{max,PV}$ = 4, $x_{min,SOM}$ = 0, $x_{max,SOM}$ = 3). $S_E(x) = x$, and $S_{SOM}$ and $S_{PV}$ use the definitions above. $j_{E,E} = j_{SOM,SOM} = j_{PV,PV} = j_{PV,SOM} = 0$, $j_{E,SOM}$ = 0.25, $j_{SOM,E}$ = −0.25, $j_{E,PV}$ = 0.4, $j_{PV,E}$ = −1, and $j_{SOM,PV}$ = −0.1. $j_{ext,PV}$ = 1.5, $j_{ext,SOM}$ = 1. Tone inputs are the same as described above.

## Acknowledgements

This work was supported by NIH R03DC013660, NIH R01DC014479, Klingenstein Award in Neuroscience, Human Frontier in Science Foundation Young Investigator Award, and the Pennsylvania Lions Club Hearing Research Fellowship to MNG and NARSAD Young Investigator Award to MA. RGN is supported by NIH NIMH T32MH017168. JJB is supported by NSF PHY1058202 and US-Israel BSF 2011058. EMG is partially supported by NINDS K12 NS049453 and the Burroughs Wellcome Fund Career Award for Medical Scientists. MNG is the recipient of the Burroughs Wellcome Fund Award at the Scientific Interface. EMG is the recipient of the Burroughs Wellcome Fund Career Award for Medical Scientists. The authors thank Jennifer Blackwell, Lisa Liu, Anh Nguyen, Daniel Feingold, Danielle Mohabir, Joshua Margolis, and Andrew Chen for technical support with experiments; Vijay Balasubramanian, Yale Cohen, Steven Eliades, Diego Contreras, Andrea Hasenstaub, and the members of the Geffen laboratory for insightful discussions and helpful comments; and Karl Deisseroth for providing the MTA for the use of some of the viruses.

## Additional information

### Funding

| Funder | Grant reference | Author |
| --- | --- | --- |
| National Institutes of Health (NIH) | R01DC014479 | Maria Neimark Geffen |
| National Institutes of Health (NIH) | R03DC013660 | Maria Neimark Geffen |
| Esther A. and Joseph Klingenstein Fund | Award in Neuroscience | Maria Neimark Geffen |
| Human Frontier Science Program (HFSP) | Young Investigator Award | Maria Neimark Geffen |
| Brain and Behavior Research Foundation | NARSAD Young Investigator Award | Mark Aizenberg |
| Pennsylvania Lions Club | Research Fellowship | Maria Neimark Geffen |
| National Institutes of Health (NIH) | NINDS K12NS049453 | Ethan M Goldberg |
| National Institutes of Health (NIH) | NIMH T32MH017168 | Ryan G Natan |
| United States-Israel Brain Science Foundation | 2011058 | John J Briguglio |
| National Science Foundation (NSF) | PHY1058202 | John J Briguglio |
| Burroughs Wellcome Fund (BWF) | Career Award at the Scientific Interface | Maria Neimark Geffen |
| Burroughs Wellcome Fund (BWF) | Career Award for Medical Scientists | Ethan M Goldberg |

The funders had no role in study design, data collection and interpretation, or the decision to submit the work for publication.

### Author contributions

RGN, MNG, Conception and design, Acquisition of data, Analysis and interpretation of data, Drafting or revising the article; JJB, Analysis and interpretation of data, Drafting or revising the article; LM-T, SIJ, MA, EMG, Acquisition of data, Analysis and interpretation of data

### Author ORCIDs

Ryan G Natan, http://orcid.org/0000-0002-3123-4833
Maria Neimark Geffen, http://orcid.org/0000-0003-3022-2993

### Ethics

Animal experimentation: All experimental procedures are in accordance with NIH guidelines and approved by the IACUC at University of Pennsylvania (protocol number 803266).

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
