## [Decision Letter]

Thank you for submitting your work entitled “Complementary control of sensory adaptation by two types of cortical interneurons” for peer review at *eLife*. Your submission has been favorably evaluated by Gary Westbrook (Senior editor), and three reviewers, one of whom is a member of the Board of Reviewing Editors.

The following individuals responsible for the peer review of your submission have agreed to reveal their identity: Andrew King (Reviewing editor); Jennifer Linden (peer reviewer); Ben Willmore (peer reviewer).

The reviewers have discussed the reviews with one another and the Reviewing editor has drafted this decision to help you prepare a revised submission.

Summary:

This paper investigates the contribution of two types of inhibitory interneuron to the generation of stimulus-specific adaptation (SSA) by neurons in the mouse auditory cortex. SSA is exhibited by neurons in non-lemniscal regions of the inferior colliculus and medial geniculate body, but appears to be a largely emergent property in the primary auditory cortex. Consequently, investigating the cortical circuitry that gives rise to this phenomenon is an important and largely unexplored question.

The authors show that optogenetic suppression of parvalbumin-positive (PV) cells produces a non-specific reduction of inhibition, which augments cortical responses to both deviant and standard tones by a similar amount (but produces an apparent reduction in the conventional “SSA index”, because the increase in response is proportionally greater for standard than for deviant tones). More interestingly, this work also shows that optogenetic suppression of somatostatin-positive (SOM) cortical interneurons specifically augments cortical responses to standard over deviant tones, and therefore reduces SSA. These results therefore suggest that PV cells contribute to SSA only through non-specific inhibition of responses to both deviant and standard tones, but that SOM cells selectively reduce cortical responses to frequently repeated standard tones and specifically amplify SSA.

The authors provide a large number of important and worthwhile controls to rule out alternative explanations for their findings, and also include a mathematical model to support their hypothesis that differences in the impact of PV and SOM inhibition on SSA could arise if the two interneuron types operate in different ranges of an essentially sigmoidal synaptic transfer function. The reviewers agree that this study provides a valuable insight into the mechanisms of SSA, but have raised a number of concerns about the analysis, modeling and presentation of the results.

Essential revisions:

1) Low impedance linear probes with recording sites separated by 50 µm were used. In a number of places, the authors refer to single unit activity. It is unlikely that the authors would have been able to isolate single units with low-impedance Neuronexus probes. At very least, more information about spike sorting and sorting quality, as well as the percentage of recordings that were actually multi-unit clusters need to be provided. The results of the present experiments are likely to be robust even if the proportion of multi-unit recordings is high, but this needs to be made clear in the manuscript (for example, for interpretation of the data in Figures 4 and 6).

2) Most of the figures show normalized firing rate measures, and most of the reported analyses were conducted on normalized firing rates. The Methods section indicates that firing rates were normalized for each neuron by dividing by the *maximum* response evoked by a deviant tone. Response measures based on a maximum (or minimum) are extremely sensitive to noise. It is very important to confirm that none of the key results would have been different had the authors used a more noise-robust normalization strategy, such as normalizing by the average response to deviant tones rather than the maximum.

3) The statistics in this paper are poorly presented, which undermines confidence in the results. Details of the statistics used are missing throughout, and need to be presented in a conventional format. The paragraph on “Statistical tests” in the Methods makes a general statement, but does not clarify what was done in individual cases. Multiple t-tests appear to be have been over used. In most places it would have been better to use a non-parametric test, such as the Wilcoxon tests that are referred to in 3 places. Where the use of multiple t-tests can be justified, it is essential to state when one- or two-tailed tests were used, and to provide effect sizes and indicate when p-values were corrected for multiple comparisons. Faith in the authors' statistical analysis is not helped by the fact that greater-than (>) signs appear to have been used where less-than (<) signs were intended in many cases (e.g. p>0.05).

4) The known impact of isofluorane and other volatile anesthetics on temporal aspects of neuronal responses – as demonstrated in rodent barrel cortex – could be a significant issue for SSA measurements Published data in cats suggests that isoflurane suppresses responses, increases latencies and increases thresholds (16). Whilst acknowledging that the effects of modulating the activity of two different inhibitory interneuron populations were studied under the same anesthetic conditions, this issue should be mentioned in the Discussion.

5) It appears that the parameters of the mathematical model (e.g. for synaptic transfer functions for simulated PV and SOM cells) were not taken from experimental data, but rather were chosen to ensure that model replicated as closely as possible the impact of PV and SOM manipulations on SSA. There is a reference in the third paragraph of the Discussion, to direct estimation of the synaptic transfer function for PV and SOM cells, but it is not evident what data were used for this estimation (and the referenced insets in Figure 7 have no quantitative axis labels). Justification for the parameters and sigmoid functions used needs to be provided. Also the model assigns equivalent connectivity patterns between PV, SOM and excitatory neurons. Although both types of interneuron receive local excitatory inputs, only PV interneurons receive thalamocortical and feedforward corticocortical inputs and PV neurons provide largely unidirectional inhibition of SOM neurons. This should be taken into account in the model. At the very least, the authors should discuss this issue as a major limitation of the model (along with the fact that there is no attempt made to model differences in the location of PV versus SOM synaptic connections of the proximal versus distal parts of the excitatory cell's dendritic tree). While it is interesting that differences in the operating ranges for PV versus SOM synaptic transfer functions might in principle explain the observed differences in their impact on SSA, the model does not “reveal” mechanisms by which the different interneuron types exert their effects – and could not do so unless the underlying parameter choices were independently validated experimentally. We are not recommending that the modelling should be removed completely, but this section of the paper is currently rather weak.

6) The way the paper is written implies that the origin of SSA in the cortex is explained here (e.g. first sentence of Discussion). In fact, however, it seems that the explanation presented here depends on SSA already being present in excitatory, SOM and PV neurons. This paper is therefore really more about how PV/SOM neurons amplify SSA effects, rather than the origin of SSA itself. This needs to be clarified throughout. In particular, the PV interneurons appear to provide a general amplification of inhibition, rather than having any role that is specific to SSA.

7) The authors show using the expression of chanelrhodopsin in cre-PV and cre-SOM mice that both types of interneuron exhibit SSA. This is an important result, but are there any differences in the SSA exhibited by these interneurons and by cortical excitatory neurons? The hypothesis put forward in the Introduction that SOM neurons show facilitation to repetitive stimulation – and are therefore likely to have a greater suppressive effect on the response of excitatory neurons to the frequently presented standard tones – implies that this effect would be mediated at the level of the synaptic inputs to the SOM neurons. Consequently, SOM neurons should respond differently from PV neurons to repeated stimulation. Moreover, if, as suggested in the subsection “PVs and SOMs differentially suppress putative excitatory neuron responses to standard and deviant tones”, SOM neurons are firing at saturation to deviant tones, the authors should have the data to show this. It would strengthen the paper if the authors could show that SOM neurons eventually facilitate with repetitive stimulation, and that SOM cell responses are saturated for deviant tones.

---

## [Author Response]

We thank the reviewers and the editors for their careful reading, thoughtful comments and detailed critiques of our manuscript. We were happy to hear that the reviewers gave an overall positive evaluation to our work. We extensively revised the manuscript, including several new analyses, as well as edits in the text. The new analyses resulted in additional supplementary figures (Figure 4—figure supplement 8, Figure 6—figure supplement 2, Figure 7—figure supplement 2). We believe that the results of these analyses and changes to the text not only address in full every concern of the reviewers but also contribute to an improved manuscript.

We first summarize the revisions, and then address in detail every comment of the reviewers. Please not that we rearranged the order in which we answer essential revisions 5-7 in order to avoid repetition.

Summary of revisions:

1) We added details to the Methods, clarifying statistical analyses and spike sorting procedures. We also now provide additional details in the text for each statistical test that we used.

2) We tested that normalizing the responses of neurons by the mean, rather than the maximum firing rate, did not affect our conclusions (Figure 4—figure supplement 8).

3) We added discussion of potential effects of anesthesia.

4) We included a discussion of the justification for including the model of excitatory-inhibitory network as a proof-of-principle test for how interneurons, which themselves adapt, differentially amplify adaptation in excitatory populations. We also added more physiologically accurate details to the model, and verified that it could still support our experimental results (Figure 7, Figure 7—figure supplement 1 and Figure 7—figure supplement 2).

5) We rephrased our interpretation of our findings to emphasize that PVs and SOMs amplify, rather than generate, SSA which is already present in both excitatory and inhibitory neurons. We further analyzed the responses of PVs and SOMs to the equal, as well as the oddball stimulus, but found no evidence for saturation of SOM responses to the deviant (Figure 6—figure supplement 2).

Essential revisions:

*1) Low impedance linear probes with recording sites separated by 50 µm were used. In a number of places, the authors refer to single unit activity. It is unlikely that the authors would have been able to isolate single units with low-impedance Neuronexus probes. At very least, more information about spike sorting and sorting quality, as well as the percentage of recordings that were actually multi-unit clusters need to be provided. The results of the present experiments are likely to be robust even if the proportion of multi-unit recordings is high, but this needs to be made clear in the manuscript (for example, for interpretation of the data in*
Figures 4 and 6*).*

We apologize for providing few details on spike sorting. We added the details in text. In particular, we did not specify the different configuration of the electrode sites on the two probes. On the Poly-2 probe, two rows of 16 electrodes each on a single shank were arranged such that each electrode site was 50 µm away from all closest neighbors. This arrangement allowed us to record densely across depth, i.e. one electrode for every 25 µm in depth. On the triode probe, electrodes were arranged in groups of 3 equidistant sites (25 µm apart). The triodes were separated vertically by 91 µm center-to-center distance, spanning 1 mm, with single sites on each end. Spike sorting was performed using commercial software (Offline Sorter, Plexon)(13). In order to improve isolation of single units from recordings using low-impedance probes, spiking activity was sorted across three (Triode, separation 25 µm) or four (Poly-2, separation 50 µm) adjacent electrode sites (67; 68). We used a stringent set of criteria to isolate single units from multiunit clusters (69; 9; 11; 26; 13; 74; 14), following published methods from our and other groups. Single-unit clusters contained <1% of spikes within a 1.0-ms interspike interval, and the spike waveforms across 3 or 4 channels had to form a visually identifiable distinct cluster in a projection onto a three-dimensional subspace.

Methods now state:

“After drilling a craniotomy and creating a durotomy exposing auditory cortex, a sillicon multi-channel probe (A1x32-Poly2-5mm-50s-177[Poly-2] or A1x32-tri-5mm-91-121-A32 [Triode], Neuronexus) was slowly lowered to between 750 µm and 1 mm into the cortex, perpendicular to the cortical surface and used to record electrical activity. Raw signals from 32 channels were bandpass filtered at 600-6000 Hz and thresholded for spike analysis, or at 10-300 Hz for local field potential (LFP) and current-source density (CSD) analysis (Poly-2 probe only), digitized at 32 kHz and stored for offline analysis (Neuralynx). Common-mode noise was removed by referencing a probe inserted in the brain outside the auditory cortex. On the Poly-2 probe, two rows of 16 electrodes each on a single shank were arranged such that each electrode site was 50 µm away from all three closest neighbors. This arrangement allowed us to record densely across depth, i.e. one electrode for every 25 µm in depth. On the triode, electrodes were arranged in groups of 3 equidistant sites, forming an equilateral triangle (25 µm separation). The triodes were separated vertically by 91 µm center-to-center distance, spanning 1 mm, with single sites on each end.

Unit identification. Spike sorting was performed using commercial software (Offline Sorter, Plexon)(13). In order to improve isolation of single units from recordings using low-impedance probes, spiking activity was sorted across three (Triode, 25 µm separation) or four (Poly-2, 50 µm separation) adjacent electrode sites (67; 68). We used a stringent set of criteria to isolate single units from multiunit clusters (69; 9; 11; 26; 13; 74; 14). Single-unit clusters contained <1% of spikes within a 1.0-ms interspike interval, and the spike waveforms across 3 or 4 channels had to form a visually identifiable distinct cluster in a projection onto a three-dimensional subspace. Putative excitatory neurons were identified based on their expected response patterns to sounds and the lack of significant suppression of the spontaneous FR due to light (53; 63). While this subpopulation may still contain inhibitory neurons, only 2% of all recorded neurons were significantly photo-suppressed at baseline (one-sided paired t-test, significance taken at p<0.05). The low impedance of the extracellular probes precluded us from conducting a more detailed analysis of cortical subpopulations based on the spike waveform (6; 63).”

2) Most of the figures show normalized firing rate measures, and most of the reported analyses were conducted on normalized firing rates. The Methods section indicates that firing rates were normalized for each neuron by dividing by the *maximum* response evoked by a deviant tone. Response measures based on a maximum (or minimum) are extremely sensitive to noise. It is very important to confirm that none of the key results would have been different had the authors used a more noise-robust normalization strategy, such as normalizing by the average response to deviant tones rather than the maximum.

This is an important point. Our rationale for using the peak response for normalization of the data when the measurements are combined across neurons was to maximize the dynamic range of representation. However, the reviewer makes an important point that potentially, such analysis might be less robust to noise than using the mean firing rate. To double-check that the results would not change quantitatively if we used a different normalization procedure, we recomputed the statistics for the key results (presented in Figure 4) based on normalization by the mean, rather than maximum response. The results are presented in Figure 4—figure supplement 8 and confirm the findings done using the original normalization analysis. We find that suppression of PVs leads to a similar increase in firing rate in response to deviants and standards, whereas suppression of SOMs leads to a selective increase in response to the standard, but not the deviant. This additional analysis demonstrates that our results do not depend on the specific aspect of normalization in averaging the data across neurons.

Methods now state:

“FR normalization was carried out separately for each tone, A and B, for each neuron by dividing the response under all conditions by the maximum FR (across 5ms bins) of the deviant tone, light-off condition. Performing this normalization by dividing response in all conditions by the mean, rather than maximum FR of the deviant tone, light-off condition did not alter significant results (Figure 4—figure supplement 8).”

3) The statistics in this paper are poorly presented, which undermines confidence in the results. Details of the statistics used are missing throughout, and need to be presented in a conventional format. The paragraph on “Statistical tests” in the Methods makes a general statement, but does not clarify what was done in individual cases. Multiple t-tests appear to be have been over used. In most places it would have been better to use a non-parametric test, such as the Wilcoxon tests that are referred to in 3 places. Where the use of multiple t-tests can be justified, it is essential to state when one- or two-tailed tests were used, and to provide effect sizes and indicate when p-values were corrected for multiple comparisons. Faith in the authors' statistical analysis is not helped by the fact that greater-than (>) signs appear to have been used where less-than (<) signs were intended in many cases (e.g. p>0.05).

We double-checked and adjusted, if necessary, all statistical tests in the manuscript. T-tests were used for single sample, paired and unpaired comparisons of data with N > 30, or when tested for normality at p< 0.05 (using Kolmogorov-Smirnov test). Bonferroni correction, which is stringent, was used for multiple comparisons. For N < 30, when the data did not meet the conditions for normality, we used the Wilcoxon Signed Rank test for single sample and paired comparisons and the Wilcoxon Rank Sum test for unpaired comparisons.

We now include an extended presentation of statistical tests. For each statistical test, we now report percent change, test used, p-value, whether one- or two-tailed test was used, z-statistic or t-statistic and number of free parameters, e.g. “..., resulting in a significant reduction in SSA index across the population (Figure 3, PV-Cre: Δ = -41%, p1 = 1e-12, t(66) = 8.6. SOM-Cre: Δ = -25%, p1 = 2e-6, t(42) = 5.4).” We use the abbreviation p1 when one-sided test was used, and p2 when two-sided test was used.

Methods now state:

“Statistical tests. For all statistical tests in which N>=30, we applied the student’s t-test (Matlab) unless specified otherwise, and reported the p-value, degrees of freedom and t-statistic. For all tests with N < 30, sample variance was tested for normality using the Komogorov-Smirnov test. If any group’s variance was non-normal, we applied a non-parametric test, e.g. Wilcoxon sign rank or rank sum test (Matlab), and provided the z-statistic for any group with a normal distribution. For all tests, Bonferroni correction was applied for multiple comparisons, and reported as “C=X” where X is the factor by which the p-value was adjusted. Statistical tests were single-tailed if there was a reasonable prior expectation about the direction of the difference between samples. p1 refers to one-sided, and p2 refers to two-sided statistics set. In all figures, single, double and triple stars indicate p < 0.05, 0.01 and 0.001 respectively. Error bars in all figures represent the standard error of the mean, unless otherwise noted.”

*4) The known impact of isofluorane and other volatile anesthetics on temporal aspects of neuronal responses – as demonstrated in rodent barrel cortex – could be a significant issue for SSA measurements Published data in cats suggests that isoflurane suppresses responses, increases latencies and increases thresholds (*[16]*). Whilst acknowledging that the effects of modulating the activity of two different inhibitory interneuron populations were studied under the same anesthetic conditions, this issue should be mentioned in the Discussion.*

We added discussion of the effect of anesthetics on auditory responses in the Discussion:

“One must be cautious in translating the data from our experiments as a strict description of neuronal activity in awake animals, as our results were based on recordings from mice under light isoflurane anesthesia. Other forms of anesthesia, such as pentobarbital-based (16; 33), ketamine (69) and high concentrations of isoflurane (16; 86), can affect multiple aspects of sound-evoked responses in the auditory cortex. Nonetheless, our results are likely to extend for awake mice, since isoflurane anesthesia-induced effects on neuronal activity decrease as the concentration of isoflurane is reduced to the levels used in our recordings (50). In addition, all recordings and manipulations were performed under identical anesthetic conditions, and our conclusions are based on the relative comparison of the effects of suppressing PVs and SOMs, which are expected to hold under awake conditions (15).”

7) The authors show using the expression of chanelrhodopsin in cre-PV and cre-SOM mice that both types of interneuron exhibit SSA. This is an important result, but are there any differences in the SSA exhibited by these interneurons and by cortical excitatory neurons? The hypothesis put forward in the Introduction that SOM neurons show facilitation to repetitive stimulation – and are therefore likely to have a greater suppressive effect on the response of excitatory neurons to the frequently presented standard tones – implies that this effect would be mediated at the level of the synaptic inputs to the SOM neurons. Consequently, SOM neurons should respond differently from PV neurons to repeated stimulation. Moreover, if, as suggested in the subsection “PVs and SOMs differentially suppress putative excitatory neuron responses to standard and deviant tones“, SOM neurons are firing at saturation to deviant tones, the authors should have the data to show this. It would strengthen the paper if the authors could show that SOM neurons eventually facilitate with repetitive stimulation, and that SOM cell responses are saturated for deviant tones.

One of our original interpretations of the differential effect of suppressing SOMs and PVs on SSA in putative excitatory neurons is that SOMs facilitated with repeated stimulus presentation, whereas PVs did not. However, when we examined the responses of optogenetically identified PVs and SOMs to the oddball stimulus, we were surprised to find that both types of interneurons themselves exhibited SSA, which was not consistent with this prediction. Rather, these results implied that the transformation may be achieved at the level of post-synaptic integration of inputs from PVs or SOMs rather than at the level of interneuron firing. Our model (Figure 7) was developed as a proof-of-principle, in order to understand whether such transformation was possible. Indeed, our model confirmed that an excitatory-inhibitory network with the properties for the non-linear transformation that we identify in Figure 7 can account for the observed effects: differential amplification of SSA by interneurons, which themselves exhibit SSA, subject to the constraints in the non-linear transfer function that we discuss.

We edited the Introduction to state:

“ We hypothesized that the two most common types of interneurons in the cortex, parvalbumin- (PVs) and somatostatin-positive cells (SOMs) (94; 76), facilitate SSA in excitatory neurons of A1 in a complementary fashion. PVs, a subset of which receive direct thalamic inputs(80), may amplify SSA in excitatory neurons by providing a constant inhibitory drive; equally strong inhibitory drive would attenuate the weak response to standard tones relatively more than the strong response to deviant tones, leading to a greater differential between standard versus deviant tone spiking response. SOMs, which target distal dendrites of pyramidal cells(60; 35), have excitatory synapses that exhibit facilitation upon repetitive stimulation(75; 78). Therefore, inputs from SOMs may exert a stimulus-specific increase in suppression of excitatory neurons that is selective to the standard and does not generalize to the deviant. Alternatively, they may contribute to selective adaptation in excitatory neurons through differential post-synaptic integration.”

In addition, we now conducted further analyses to test whether there were differences in adaptation between PVs and SOMs by examining their responses to both the oddball (in which tones A and B were presented at 90-10 or 10-90 ratio) and the “equal” stimulus (in which tones A and B were presented at 50-50 ratio). We found some significant differences that are potentially of interest, but that still do not provide evidence for saturation at the level of Somatostatin outputs (Figure 6—figure supplement 2): The equal stimulus does not drive adaptation in PVs, whereas it drives significant adaptation in SOMs, which suggests that initial adaptation to repeated tones in SOMs occurs on a faster time scale than in PVs. For both PVs and SOMs, the response to the standard tone is significantly lower than response to the equal stimulus, providing evidence for an additional time scale of adaptation.

*5) It appears that the parameters of the mathematical model (e.g. for synaptic transfer functions for simulated PV and SOM cells) were not taken from experimental data, but rather were chosen to ensure that model replicated as closely as possible the impact of PV and SOM manipulations on SSA. There is a reference in the third paragraph of the Discussion, to direct estimation of the synaptic transfer function for PV and SOM cells, but it is not evident what data were used for this estimation (and the referenced insets in*
Figure 7
*have no quantitative axis labels). Justification for the parameters and sigmoid functions used needs to be provided. Also the model assigns equivalent connectivity patterns between PV, SOM and excitatory neurons. Although both types of interneuron receive local excitatory inputs, only PV interneurons receive thalamocortical and feedforward corticocortical inputs and PV neurons provide largely unidirectional inhibition of SOM neurons. This should be taken into account in the model. At the very least, the authors should discuss this issue as a major limitation of the model (along with the fact that there is no attempt made to model differences in the location of PV versus SOM synaptic connections of the proximal versus distal parts of the excitatory cell's dendritic tree). While it is interesting that differences in the operating ranges for PV versus SOM synaptic transfer functions might in principle explain the observed differences in their impact on SSA, the model does not “reveal” mechanisms by which the different interneuron types exert their effects – and could not do so unless the underlying parameter choices were independently validated experimentally. We are not recommending that the modelling should be removed completely, but this section of the paper is currently rather weak.*

As explained above, the model was constructed as a proof-of-principle for understanding our experimental findings, rather than with the goal of replicating a detailed biological network. Therefore, the parameters, including the strength of synaptic constants and the details of the transfer function, were empirically chosen to achieve a good match between model output and experimental data.

The reviewers draw on two important observations that we did not include in the original model: (1) SOMs do not seem receive direct thalamic inputs, whereas PVs do; and (2) PVs and SOMs may inhibit each other. The connectivity between PVs and SOMs is highly complex, and there are controversial findings on functionality of the connections, which is why we opted in the original version of the manuscript to start with examining the simplest possible excitatory-inhibitory network without including details about interneuron connectivity. For instance, some papers have found that SOMs do receive inputs directly from the thalamus (84). Furthermore, while the reviewers suggest that there is inhibition from PVs to SOMs, several recent studies find that PVs do not contribute to inhibition of SOMs, whereas SOMs provide inhibition to PVs (19; 72; 81).

In the revision, we examined the effect of incorporating these connections in the model by making the following modifications:

A) We modified the model such that, compared to PVs, SOMs received delayed tone-evoked inputs as would be expected if there were at least one synapse in-between thalamic inputs and SOMs – Figure 7 has now been adjusted with the new parameters. Incorporating this delay did not affect the results.

B) We incorporated inhibition from SOMs to PVs in the model and tested whether the prediction for differential effect of SOMs and PVs in SSA still held. We found that the model with added inhibitory inputs from SOMs to PVs could still replicate the experimental results. This figure is now included as Figure 7—figure supplement 2.

Results presented in Figure 7 and Figure 7—figure supplement 1 and Figure 7—figure supplement 2 demonstrate that the excitatory-inhibitory model still supports our experimental results, after these adjustments. Furthermore, our manuscript now discusses that the model serves as a proof-of-principle to verify the interpretation of our results, rather than to reveal a specific mechanism.

Results now state: “Excitatory and inhibitory neurons form tight mutually coupled networks in A1, and we hypothesized that through differential post-synaptic integration by excitatory neurons, interneurons can amplify adaptation in excitatory neurons. As a proof-of-principle that would account for our findings that PVs and SOMs exhibit similar magnitude of SSA, yet have a differential effect on SSA in putative excitatory neurons, we constructed a simplified model of mutually coupled inhibitory-excitatory neuronal populations. ... SOMs have been shown to inhibit PVs (19; 72; 81). Including inhibition between SOMs and PVs did not affect the model outcome, with suppression of PVs resulting in suppression of excitatory responses to both the standard and the deviant, and suppression of SOMs driving specific suppression of excitatory responses to the standard, but not the deviant (Figure 7—figure supplement 2).”

6) The way the paper is written implies that the origin of SSA in the cortex is explained here (e.g. first sentence of Discussion). In fact, however, it seems that the explanation presented here depends on SSA already being present in excitatory, SOM and PV neurons. This paper is therefore really more about how PV/SOM neurons amplify SSA effects, rather than the origin of SSA itself. This needs to be clarified throughout. In particular, the PV interneurons appear to provide a general amplification of inhibition, rather than having any role that is specific to SSA.

We corrected the Discussion throughout to emphasize that interneurons serve to amplify SSA, which may already be present upon integration of thalamo-cortical inputs.

Abstract: “We found that two types of cortical interneurons differentially amplify SSA in putative excitatory neurons… A mutually coupled excitatory-inhibitory network model accounts for the distinct mechanisms by which cortical inhibitory neurons enhance sensitivity to unexpected sounds.”

Impact Statement: “We discover that two distinct types of inhibitory neurons increase the brain’s sensitivity to unexpected acoustic signals, by amplifying selective suppression of cortical responses to frequent, but not rare sounds.”

Introduction: “Therefore, cortical circuits are proposed to contribute to and amplify SSA in A1… We found that both types of interneurons contribute to SSA in the cortex, with PVs providing constant inhibition, and SOMs increasing their effect with repeated tones.”

Discussion: “Across sensory modalities, cortical neurons exhibit adaptation, attenuating their responses to redundant stimuli(20; 88; 34; 4; 47). Adaptation to stimulus context is thought to increase efficiency of sensory coding under the constraints of limited resources(5). Yet the neuronal-circuit mechanisms that facilitate adaptation in the cortex remain poorly understood. In the primary auditory cortex (A1), the vast majority of neurons exhibit stimulus-specific adaptation (SSA, Figure 1).*”*